# ProConMV: Provenance-Enabled Conceptual Framework for Interpretable Multi-View Diabetic Retinopathy Diagnosis

**Xiaoling Luo** [1]  **Shuo Yang** [1]  **Qihao Xu** [2]  **Jiansong Zhang** [1]  **Zhuoqin Yang** [1]  **Zhihui Lai** [1]  **Linlin Shen** [3]
**Chengliang Liu** [4]

## Abstract

Existing deep learning models have demonstrated potential in Diabetic retinopathy (DR) diagnosis, but they still suffer from three key challenges: reliance on single-source inputs, opaque and untraceable reasoning processes, and the absence of a mechanism for result verification. Thus, we propose a provenance-enabled concept-based framework for multi-view DR diagnostic (ProConMV), which integrates DR lesion masks, clinical text and multi-view data, utilizing multimodal prompt analysis and visual-text concept interaction to learn the interpretable multi-source input. During the reasoning stage, the proposed framework introduces lesion concepts for causal reasoning chains combining clinical guidelines, and adds doctor intervention for human-machine collaboration. For dynamic fusion decision and verification in multi-view DR diagnosis, we derive via generalization theory that incorporating each view's lesion concept uncertainty and grading uncertainty reduces the generalization error upper bound. Accordingly, we design a dual uncertainty-aware module to enable provenance-based verification, ultimately enabling verifiable analysis of DR diagnostic results. Extensive experiments conducted on two public multi-view DR datasets demonstrate the effectiveness of our method. The code will be released at https://github.com/SoY0ung/ProConMV.

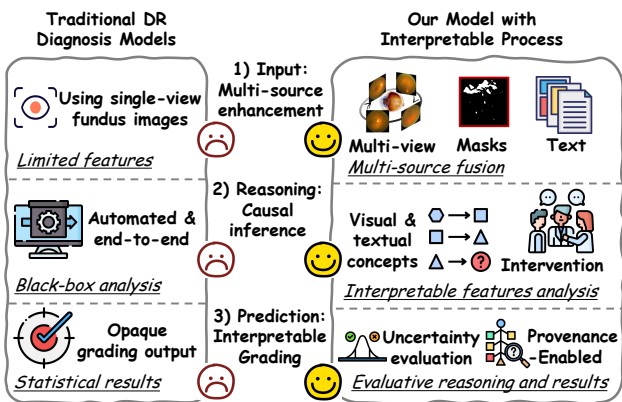

*Figure 1.* Our model with an interpretable process compared with traditional DR diagnosis models.

## 1. Introduction

Diabetic retinopathy (DR) is a major cause of blindness among diabetic patients (Federation, 2025), posing a visual health error to the global working-age population. International DR severity is diagnosed by lesions like microaneurysms (MA), hemorrhage (HE), and exudation (EX), and classified into five grades (Grade 0-4): normal, mild, moderate, severe, and Proliferative Diabetic Retinopathy (PDR) (Wilkinson et al., 2003). With the development of artificial intelligence technology, traditional deep learning models (Liu et al., 2022a; 2025) have demonstrated excellent performance in DR grading tasks, capable of quickly processing large amounts of images and providing grading results. However, their inherent limitations in practical application have gradually become bottlenecks in bridging the gap between AI technology and real-world medical needs.

A critical examination of existing DR diagnostic models reveals three core challenges that undermine their credibility and usability in clinical settings (Lin et al., 2025), as illustrated in Fig.1. First, single-source input limitations persist: most models rely solely on monomodal data and fail to integrate complementary information from lesion morphology and clinical text. Moreover, training on single-view databases (Decenciere et al., 2014; EyePACS, 2015) means the field of view (FOV) of input images covers only 20% of

---

[1]College of Computer Science and Software Engineering, Shenzhen University, China [2]Shenzhen Key Laboratory of Visual Object Detection and Recognition, Harbin Institute of Technology, Shenzhen, China [3]School of Artificial Intelligence, Shenzhen University, China [4]Department of Computer and Information Science, University of Macau, Macau. Correspondence to: Qihao Xu <xqh51199597@outlook.com>.

*Proceedings of the 43rd International Conference on Machine Learning*, Seoul, South Korea. PMLR 306, 2026. Copyright 2026 by the author(s).

the observable fundus, increasing the error of missing critical pathological features. Second, "black-box" reasoning processes lack medical interpretability (Huang et al., 2024): the internal calculations of traditional models are opaque, and they cannot map image features to diagnostic results via clinically understandable logic. Third, insufficient result verification mechanisms (Luo et al., 2025): existing methods generally lack uncertainty quantification and traceable validation, making it impossible to assess the reliability of outputs. This deficiency is problematic in medical scenarios, where unreliable results may lead to misdiagnosis, missed diagnosis, or inappropriate clinical interventions.

To address these critical issues and alleviate the credibility dilemma of DR diagnosis models in clinical practice, this study proposes a **Pro**venance-enabled **Con**ceptual framework for **M**ulti-**V**iew diabetic retinopathy diagnosis (**ProConMV**), encompassing multi-source input fusion, interpretable causal reasoning, and verifiable result evaluation. Specifically, we integrate fundus image lesion masks, structured clinical texts, and multi-view fundus data to construct a rich input space. The proposed Hilbert RWKV encodes spatial features of images for precise lesion localization, while a large language model (LLM)-based text encoder (Singh et al., 2025) extracts lesion-related semantic information from doctor-verified clinical texts, with cross-modal interaction enabled by a Visual-Text RWKV (VT-RWKV) module. For reasoning, we introduce lesion concepts (Wen et al., 2024) as intermediate units aligned with clinical guidelines. And incorporate real-time doctor intervention to build a human-machine collaborative causal reasoning chain, transforming input-output mapping into physician-understandable pathological logic.

Furthermore, multi-view fusion decision-making is crucial for the comprehensive DR diagnosis. However, due to varying cooperation among different patients during fundus examinations, the captured multi-view fundus images exhibit various variations. Most existing multi-view fusion methods (Xue & Marculescu, 2023; Cao et al., 2024; Xin et al., 2025) lack theoretical guarantees, which can lead to one-sided and inaccurate diagnostic results. To achieve reliable dynamic fusion, we demonstrate for the first time in a multi-view concept-based model that, from the perspective of generalization theory, when fusion weights are negatively correlated with both concept loss and grading loss, the upper bound of the generalization error for decision fusion will be reduced and outperforms that of static fusion methods. Meanwhile, the concept uncertainty and grading uncertainty of each view related to the decision are traceable, enabling verifiable analysis of DR diagnostic results. The main contributions of our ProConMV are summarized as follows:

- The multimodal input mechanism is proposed to integrate DR lesion masks, clinical text, and multi-view data. Leveraging Hilbert RWKV encoding of image features and textual concept encoder extraction of text features to achieve cross-modal interaction, a semantically rich interpretable input foundation is provided for reasoning.

- A causal reasoning chain combining lesion concepts and clinical guidelines is constructed, with the simultaneous introduction of a doctor intervention link to form a human-machine collaborative reasoning mode, effectively solving the problem of opaque and untraceable reasoning processes in traditional models.

- In the dynamic fusion decision, we derive for the first time from the perspective of generalization that incorporating the lesion concept uncertainty and the grading uncertainty of each view can reduce the generalization error upper bound. Then, we design a dual uncertainty-aware module to realize provenance-enabled verification of diagnostic results.

**Conflict of Interest Disclosure.** The authors declare no financial conflicts of interest related to this work.

## 2. Related Work

### 2.1. DNN-Based Methods for Multi-View Diabetic Retinopathy Diagnosis

Recently, multi-view approaches for DR diagnosis have attracted increasing attention. Luo et al. (Luo et al., 2023) first proposed MVCINN, a multi-view DR diagnosis network that integrates CNNs and Transformers. Subsequently, several works (Luo et al., 2024; Lin et al., 2025; Hu et al., 2025) have leveraged visual cues, such as vessel and DR lesion masks derived from segmentation models, to improve diagnostic accuracy. Others (Hou et al., 2022; Luo et al., 2025) focused on inter-view information exchange and backbone design to further strengthen diagnostic representations. Nevertheless, interpretability for this task has not been adequately explored, particularly in terms of textual explanations and transparent diagnostic workflows, which are of great significance in clinical medical diagnosis.

### 2.2. Interpretable Machine Learning Models in Computer Vision

Interpretability methods have achieved remarkable success in computer vision, which enhances human understanding of model predictions. Early studies on interpretability mainly focused on post-hoc explanations of black-box models, such as Shapley (Roth, 1988; Chen et al., 2022), Grad-CAM (Selvaraju et al., 2017; Chattopadhay et al., 2018), and Prototypes (Seo et al., 2023). However, these methods lack human-comprehensible reasoning processes and are

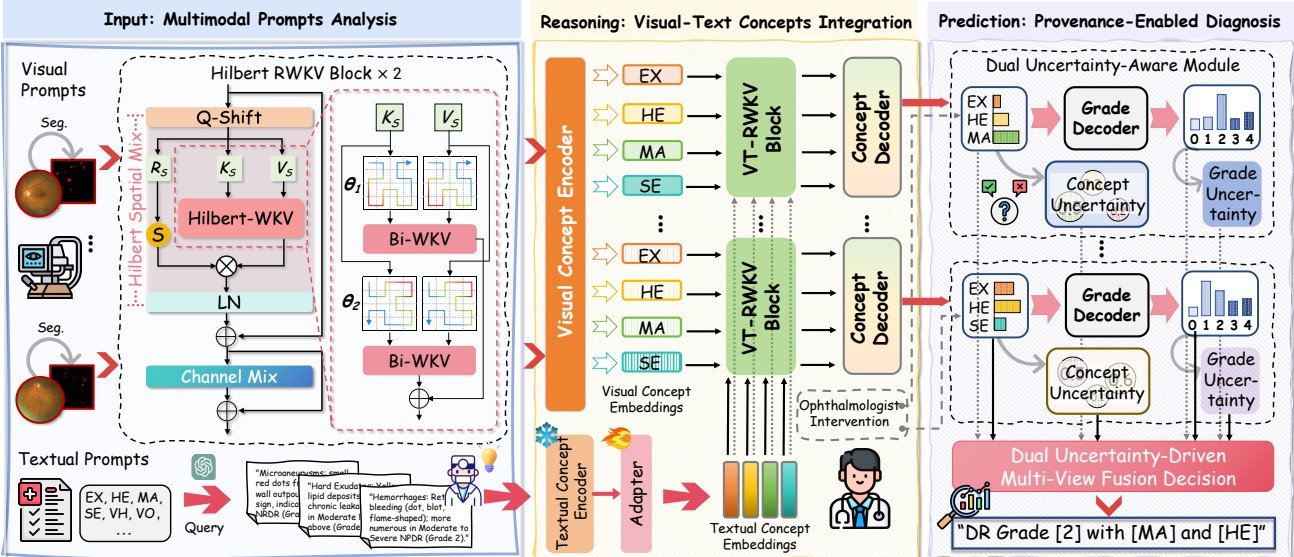

*Figure 2.* The framework of our proposed ProConMV model has three parts: multi-source input for the enhancement of interpretable features, visual-text concepts integration for causal reasoning, and provenance-enabled diagnosis using the dual uncertainty-aware module.

therefore fundamentally unable to provide reasonable explanations for downstream applications. To this end, Koh et al. (Koh et al., 2020) proposed the Concept Bottleneck Model (CBM), an interpretable framework that first predicts visual concepts and then uses them to generate the final prediction. There is a diverse set of CBM variants (Espinosa Zarlenga et al., 2022; Zhang et al., 2024; Wen et al., 2024; Ciravegna et al., 2022; Sun et al., 2025), each tackling the problem from a new perspective. To our knowledge, MVCBM (Klimiene et al., 2022b) and SSMVCBM (Marcinkevičs et al., 2024) are the only two CBM-based works on interpretable multi-view classification. These methods still have limitations in both accuracy and interpretability.

### 2.3. Receptance Weighted Key Value

Receptance Weighted Key Value (RWKV) (Peng et al., 2023) is a neural network architecture that combines the parallel training ability of Transformers with the efficient recurrence of RNNs, characterized by its linear complexity and effectiveness in modeling long sequences. Recently, RWKV has gained renewed attention in vision tasks, as its core WKV attention mechanism has demonstrated superior performance compared to self-attention (Dosovitskiy et al., 2021) in some vision domains. Duan et al. proposed Vision-RWKV (Duan et al., 2025), first introducing a bidirectional WKV attention mechanism and a quad-directional token shift method to adapt RWKV for image classification tasks. Building upon RWKV and Vision-RWKV, some variants have been introduced for diverse vision-related tasks, including RWKV-SAM (Yuan et al., 2024) for segmentation, RWKV-CLIP (Gu et al., 2024) for vision-language representation learning, and Point-RWKV (He et al., 2025) for 3D

point cloud learning. However, these works overlook the problem of spatial locality loss introduced by token serialization in image modeling. To address this, our Hilbert-RWKV backbone introduces a Hilbert-curve scanning mechanism into the RWKV architecture.

## 3. Method

This framework takes the fusion of multi-source clinical data as its foundational input, utilizes medically logical causal reasoning as its core link, and employs a dual uncertainty-aware mechanism to guarantee its results. Wherein, visual prompts are obtained from the DR lesion segmentation model (Xu et al., 2024) (pretrained on external datasets), whereas textual prompts comprise doctor-verified conceptual information generated by GPT-5 (Singh et al., 2025).

### 3.1. Multi-View Concept Representation Learning

Some studies (Xu et al., 2021; Shamshad et al., 2023; Gu et al., 2024; Liu et al., 2026; Zhang et al., 2026) have demonstrated that existing Transformer-based multi-view methods are less effective at fine-grained local concept perception, while incurring large parameter overhead and prolonged inference times. To capture multi-view fine-grained lesion concept features, we propose an RWKV-based backbone equipped with a multi-directional Hilbert attention mechanism. By leveraging the locality-preserving nature of Hilbert curves to maintain spatial proximity during serialization, this design mitigates directional bias and promotes isotropic feature aggregation while ensuring linear complexity (Lawder & King, 2000; Moon et al., 2001). Specifically, the backbone first utilizes a stem (comprising two convolu-

tional layers and downsampling) to extract shallow features for each view. Then, it optimizes the deep features using two Hilbert RWKV Blocks.

### 3.1.1. HILBERT RWKV BLOCK

This block mainly consists of two components: Hilbert spatial-mix and channel-mix. The spatial mixing is the core, while the channel mixing serves as a feed-forward network (FFN) to enhance channel features. Given the fundus visual representation of the $v$-th view $\mathbf{x}^{(v)} \in \mathbb{R}^{h \times w \times d}$, the block first transforms it into $p \times p$ patches, which are then projected into visual tokens of shape $\frac{hw}{p^2} \times d$. These tokens $\overline{\mathbf{x}}^{(v)}$ are fed into the Hilbert spatial-mix module. Similar to Vision-RWKV (Duan et al., 2025), we adopt the quad-directional token shift (Q-Shift) operation along with three parallel linear layers to obtain the matrices $\mathbf{R}_s, \mathbf{K}_s, \mathbf{V}_s \in \mathbb{R}^{\frac{hw}{p^2} \times d}$:

$$\mathbf{R}_s = \text{Q-Shift}_R(\overline{\mathbf{x}}^{(v)})W_R, \quad \mathbf{K}_s = \text{Q-Shift}_K(\overline{\mathbf{x}}^{(v)})W_K,$$
$$\mathbf{V}_s = \text{Q-Shift}_V(\overline{\mathbf{x}}^{(v)})W_V. \tag{1}$$

This Q-Shift operation enhances the attention mechanism by allowing tokens to shift and perform linear interpolation with neighboring tokens, thereby improving the receptive field of each token without increasing computational complexity. The following formula holds:

$$\text{Q-Shift}_{(*)}(\overline{\mathbf{x}}^{(v)}) = \overline{\mathbf{x}}^{(v)} + (1 - \mu(*))\overline{\mathbf{x}}'^{(v)},$$
$$\overline{\mathbf{x}}'^{(v)}[a,b] = \text{Concat}\Big(\overline{\mathbf{x}}^{(v)}[a-1,b,0{:}\tfrac{d}{4}], \ \overline{\mathbf{x}}^{(v)}[a+1,b,\tfrac{d}{4}{:}\tfrac{d}{2}],$$
$$\overline{\mathbf{x}}^{(v)}[a,b-1,\tfrac{d}{2}{:}\tfrac{3d}{4}], \ \overline{\mathbf{x}}^{(v)}[a,b+1,\tfrac{3d}{4}{:}d]\Big). \tag{2}$$

where the subscript $(*) \in \{R, K, V\}$ represents the interpolation of $\overline{\mathbf{x}}^{(v)}$ and $\overline{\mathbf{x}}'^{(v)}$, controlled by the learnable vector $\mu(*)$. Subsequently, we design a novel linear attention mechanism with local continuity perception, Hilbert-WKV($\mathbf{K}_s, \mathbf{V}_s$), and a gating function $\sigma(\mathbf{R}_s)$ to obtain the output of the Hilbert spatial mixing module $\mathbf{O}_s$, as shown in the figure:

$$\mathbf{O}_s = \text{LN}\big(\sigma(\mathbf{R}_s) \odot \text{Hilbert-WKV}(\mathbf{K}_s, \mathbf{V}_s)W_{O_s}\big). \tag{3}$$

Here, $\sigma$ represents the sigmoid function, $\odot$ denotes element-wise multiplication, and LN refers to layer normalization. To achieve channel feature fusion, $\mathbf{O}_s$ is passed into the channel-mix module. $\mathbf{R}_c, \mathbf{K}_c, \mathbf{V}_c \in \mathbb{R}^{\frac{hw}{p^2} \times d}$ are obtained similarly to spatial-mix by $\mathbf{O}_s$. In the channel-mix module, $\mathbf{V}_c$ is the linear projection of $\mathbf{K}_c$ after applying the activation function SquaredReLU, controlled by a gating mechanism $\sigma(\mathbf{R}_c)$. The output $\mathbf{O}_c$ is the linear projection of the resulting value:

$$\mathbf{O}_c = \sigma(\mathbf{R}_c) \odot (\text{SquaredReLU}(\mathbf{K}_c)W_V) W_{O_c}. \tag{4}$$

### 3.1.2. HILBERT-WKV ATTENTION MECHANISM

Inspired by the filling curve (Chen et al., 2023) and the bidirectional attention mechanism Bi-WKV (Duan et al., 2025), we design the Hilbert-WKV, a multi-directional attention mechanism grounded in the Hilbert curve. Our proposed Hilbert-WKV has two advantages in multi-view fundus representation learning. As shown in Fig. 2, it preserves the continuity of token arrangement, and the local scanning characteristic of the Hilbert curve window outperforms the default strip scanning. Specifically, after dividing into $\frac{hw}{p^2}$ tokens of size $p \times p$, the order of tokens is determined based on the 2D Hilbert curve:

$$H_n(a,b) = \begin{cases} 4 \cdot H_{n-1}(b,a), & (a,b) \in Q_0 \\ 4 \cdot H_{n-1}(a,b) + 4^{n-1}, & (a,b) \in Q_1 \\ 4 \cdot H_{n-1}(a,b) + 2 \cdot 4^{n-1}, & (a,b) \in Q_2 \\ 4 \cdot H_{n-1}(N{-}1{-}b, N{-}1{-}a) & (a,b) \in Q_3 \\ \quad + 3 \cdot 4^{n-1}. \end{cases} \tag{5}$$

Here, $H_n(a,b)$ represents the Hilbert sequence position of the token located at $(a,b)$, with $N = 2^n = \frac{1}{p}\sqrt{hw}$ and $H_n(0,0) = 0$. $Q_0$ to $Q_3$ represent the four quadrants formed by dividing the area of $N/2$ into four sections: $Q_0$ (lower-left), $Q_1$ (upper-left), $Q_2$ (upper-right), and $Q_3$ (lower-right). We denote the Hilbert Transform as $\eta$ and its inverse as $\eta^{-1}$. The proposed Hilbert-WKV attention mechanism constructs attention mechanisms with vertical and horizontal direction priorities:

$$\text{Hilbert-WKV}(\mathbf{K}_s, \mathbf{V}_s) = \tag{6}$$
$$\underbrace{\eta^{-1}(\text{Bi-WKV}(\overline{\mathbf{K}}_s, \overline{\mathbf{V}}_s))}_{\text{Vertical Attention}} + \underbrace{\eta^{-1}(\text{Bi-WKV}(\overline{\mathbf{K}}_s^{\mathsf{T}}, \overline{\mathbf{V}}_s^{\mathsf{T}}))^{\mathsf{T}}}_{\text{Horizontal Attention}},$$

where $\overline{\mathbf{K}}_s = \eta(\mathbf{K}_s)$, $\overline{\mathbf{V}}_s = \eta(\mathbf{V}_s)$, and $\mathsf{T}$ is the transpose. The Bi-WKV attention calculation for the $t$-th token is formulated as follows:

$$wkv_t = \text{Bi-WKV}(\mathbf{K}_s, \mathbf{V}_s)_t$$
$$= \frac{\sum_{i \neq t} e^{-(|t-i|-1)/T \cdot w + k_i} v_i + e^{u+k_t} v_t}{\sum_{i \neq t} e^{-(|t-i|-1)/T \cdot w + k_i} + e^{u+k_t}}, \tag{7}$$

where $T = \frac{hw}{p^2}$ represents the number of tokens. $w$ and $u$ are two $D$-dimensional learnable vectors representing channelwise spatial decay and the current token, respectively. $k_t$ and $v_t$ denote the $t$-th feature of $\mathbf{K}_s$ and $\mathbf{V}_s$. Compared to the self-attention, the Hilbert-WKV attention achieves linear complexity $O(n \times T \times D)$, where $n$ is a constant.

### 3.1.3. VISUAL CONCEPT ENCODER

Following the shared backbone processing, each view obtains its latent representation $\overline{\mathbf{h}}^{(v)} \in \mathbb{R}^{n_h}$. Our model then feeds $\overline{\mathbf{h}}^{(v)}$ into a concept-specific fully connected layer,

which learn the lesion concept embedding in $\mathbb{R}^{n_z}$, namely $\mathbf{z}_j^{(v)} = \sigma(W_j \bar{\mathbf{h}}^{(v)} + b_j)$. Here, $\mathbf{z}_j^j$ denotes the $j$-th concept embedding in the $i$-th view, while $\sigma$, $W_j$, and $b_j$ correspond to the LeakyReLU activation function, weight paremeters, and bias term of the $j$-th concept layer, which are shared across all views. In this way, the fundus visual feature is mapped into lesion concept representations for each view.

### 3.1.4. TEXTUAL CONCEPT ENCODER

We utilize GPT-5 to generate descriptions for each DR lesion concept, focusing on their characteristics and progression stages. This description text is a curated knowledge base for retinal diagnosis, which provides a unified and doctor-verified textual description for all samples as a shared semantic anchor point. It is then fed into the medical text encoder (Singh et al., 2025) and a trainable, lightweight MLP-based adapter (Gao et al., 2024) to generate the textual concept embeddings $\mathbf{t}_j \in \mathbb{R}^{n_t}$, where $j$ denotes the $j$-th concept.

### 3.2. Multi-View Visual-Text Concept Integration

The incorporation of textual information provides intuitive evidence for DR diagnosis. To efficiently align multi-view lesion concept representations with human clinical knowledge and improve reasoning interpretability, we propose a visual-text RWKV (VT-RWKV) block, a multi-model driven concept enhancement method based on RWKV.

Specially, for view $v$, our model considers the concatenated representation of the concept visual embedding $\mathbf{z}^{(v)} = [\mathbf{z}_1^{(v)}, \mathbf{z}_1^{(v)}, \ldots, \mathbf{z}_M^{(v)}]$ and its corresponding textual embedding $\mathbf{t} = [\mathbf{t}_1, \mathbf{t}_2, \ldots, \mathbf{t}_M]$ as input, where $M$ denotes the number of lesion concepts. The multi-modal embeddings are then projected through three parallel linear layers to obtain the matrices $\mathbf{R}_{con}, \mathbf{W}_{con}, \mathbf{K}_{con} \in \mathbb{R}^{m \times n_z}$:

$$\mathbf{R}_{con} = W_r \mathbf{z}^{(v)}, \quad \mathbf{K}_{con} = W_k \mathbf{t}, \quad \mathbf{V}_{con} = W_v \mathbf{t}, \quad (8)$$

where $W_r$, $W_k$, and $W_v$ are learnable parameters. Here, the VT-RWKV operator improves concept visual representations by fusing them with aligned textual features. The key and value matrices $\mathbf{K}_{con}$ and $\mathbf{V}_{con}$, computed from $\mathbf{t}$, are fed into a linear complexity bidirectional attention module, Bi-WKV, to obtain the attention output $wkv \in \mathbb{R}^{M \times n_z}$. Meanwhile, the visual embedding $\mathbf{z}^{(v)}$ generates a gating matrix $\sigma(\mathbf{R}_{con})$, which modulates the attention output. The enhanced concept representation $\bar{\mathbf{z}}^{(v)}$ is computed as:

$$\begin{aligned} \bar{\mathbf{z}}^{(v)} &= (\sigma(\mathbf{R}_{con}) \odot cwkv) W_z, \\ cwkv &= \text{Bi-WKV}(\mathbf{K}_{con}, \mathbf{V}_{con}). \end{aligned} \quad (9)$$

Where $W_z$ is a learnable projection matrix, $\sigma$ denotes the sigmoid function, and $\odot$ represents element-wise multiplication. Through this fusion, the model obtains each view's

lesion concept embeddings that are aligned with both visual information and diagnostic knowledge, thereby enhancing the interpretability and predictive accuracy of the concepts.

In reasoning, the view-shared concept decoder $C$ transforms the enhanced concept representation of each view into its corresponding concept predictions, which are then passed to the grade decoder $G$ to produce the final grading result for that view. For view $v$, the procedure can be derived as:

$$\hat{\mathbf{c}}^{(v)} = C(\bar{\mathbf{z}}^{(v)}) \in \mathbb{R}^M, \quad \hat{\mathbf{y}}^{(v)} = G(\hat{\mathbf{c}}^{(v)}) \in \mathbb{R}^K. \quad (10)$$

Here, $\hat{\mathbf{c}}^{(v)}$ denotes the concept prediction of view $v$ with $M$ concepts, and $\hat{\mathbf{y}}^{(v)}$ represents the corresponding grading vector with $K$ DR grades. In this way, the model completes the entire process from input to concept analysis and finally to grading output for each view, i.e., $\mathbf{x}^{(v)} \to \hat{\mathbf{c}}^{(v)} \to \hat{\mathbf{y}}^{(v)}$.

### 3.3. Dual Uncertainty-Aware Interpretable Multi-View DR Diagnosis

Existing multi-view late fusion methods (Han et al., 2022; Zhang et al., 2023; Xin et al., 2025) focus on the reliability of the final predictions from each view. However, these exhibit limitations within the multi-view concept reasoning pipeline. The middle lesion concepts, which serve as the input to the $\mathbf{c} \to \mathbf{g}$ stage, play a critical role in determining both the interpretability and the accuracy of the reasoning process for each view. Thus, from the perspective of generalization theory, we propose a dynamic fusion method with grading and concept dual uncertainty awareness in interpretable multi-view DR decision-making.

### 3.3.1. GENERALIZATION THEORY IN MULTI-VIEW CONCEPT-BASED MODELS

We first integrate the generalization theory into the multi-view concept-based framework. This enables us to formalize the reasoning process, analyze final grading loss, and validate weight design in the following setting and derivation.

**Setting.** In conjunction with Equ. (10), we define $\mathbf{c}^{(v)}$ and $\mathbf{y}^{(v)}$ as the concept label and the grading (class prediction) label of view $v$, respectively. According to the reasoning pipeline $\mathbf{x}^{(v)} \to \hat{\mathbf{c}}^{(v)} \to \hat{\mathbf{y}}^{(v)}$, The view-shared concept predictor $C$ and grading predictor $G$ are specified in the hypothesis spaces $\mathcal{C}$ and $\mathcal{G}$. The final prediction of the late-fusion multi-view method is formulated as $\hat{\mathbf{y}} = \sum_{v=1}^V w_v \hat{\mathbf{y}}^{(v)}$, where $w_v \in (0, 1)$ denotes the fusion weight of view $v$, satisfying $\sum_{v=1}^V w_v = 1$.

Unlike the static fusion weight $w_v^s = 1/V$, dynamic weight $w_v^d$ is dependent on the input. To provide a provable dynamic weight design for multi-view concept-based models, we introduce generalization theory. The generalization error of grading classification in multi-view concept-based models $L_y$ can be expressed as: $L_y =$

$\mathbb{E}_{(\mathbf{x}^{(1:V)}, \mathbf{c}^{(1:V)}, \mathbf{y}) \sim \mathcal{D}} \left[ \ell_y \left( \sum_{v=1}^{V} w_v \, G(\hat{\mathbf{c}}^{(v)}), \mathbf{y} \right) \right]$, where $\mathbb{E}$ is the expectation, $\mathcal{D}$ is the unknown joint distribution and $\ell_y$ represents the convexity cross-entropy loss. Our objective is to search for dynamic $w_v^d$ that minimizes the upper bound of $L_y$ as much as possible, and to prove that it is superior to the static fusion weight $w_v^s$.

**Theorem 1** (Generalization Bound of Decision Fusion in Multi-View Concept-based Models)

Given a training set $\mathcal{D}_{\text{train}} = \left\{ \left( \mathbf{x}_i^{(1:V)}, \mathbf{c}_i^{(1:V)}, \mathbf{y}_i \right) \right\}_{i=1}^{N}$, we derive the generalization error bound of multi-view concept-based models using Rademacher complexity (Bartlett & Mendelson, 2002), and for $1 > \delta > 0$, with probability at least $1 - \delta$, it holds that

$$L_y \leq \underbrace{\sum_{v=1}^{V} \mathbb{E}[w_v] \hat{L}_y^{(v)} + \sum_{v=1}^{V} \mathbb{E}[w_v] L_g^{(v)} \hat{L}_c^{(v)}}_{\text{Term-L (average empirical loss)}} + \underbrace{2P \sqrt{\frac{\ln(V/\delta)}{N}}}_{\text{concentration term}}$$

$$+ \underbrace{\sum_{v=1}^{V} \mathbb{E}[w_v] \mathfrak{R}_N(\mathcal{G}) + \sum_{v=1}^{V} \mathbb{E}[w_v] L_g^{(v)} \mathfrak{R}_N(\mathcal{C})}_{\text{Term-C (average complexity)}}$$

$$+ \underbrace{\sum_{v=1}^{V} \text{Cov}\left( w_v, \ell_y(G(\mathbf{c}^{(v)}), \mathbf{y}) \right)}_{\text{Term-Cov (grading weights \& losses)}} \quad (11)$$

$$+ \underbrace{\sum_{v=1}^{V} L_g^{(v)} \text{Cov}(w_v, \|\hat{\mathbf{c}}^{(v)} - \mathbf{c}^{(v)}\|_1)}_{\text{Term-Cov (concept weights \& losses)}}.$$

Where $\hat{L}_y^{(v)}$ and $\hat{L}_c^{(v)}$ denote the empirical prediction error (evaluated under true concepts) and the empirical concept error, respectively. $\mathfrak{R}_N(\mathcal{G})$ and $\mathfrak{R}_N(\mathcal{C})$ denote the Rademacher complexities estimated with $N$ samples, $L_g^{(v)} > 0$ is the Lipschitz constant of $G$ with respect to its concept input (i.e., the sensitivity bound of the prediction loss with respect to the concept). $\text{Cov}(\cdot, \cdot)$ denotes the covariance, and $P > 0$ is an absolute constant determined by the boundedness of the loss. In particular, when $w_v = w_v^s = 1/V$, the Term-Cov becomes 0.

First, since $\hat{L}_y^{(v)}$, $\hat{L}_c^{(v)}$, $L_g^{(v)}$, $\mathfrak{R}_N(\mathcal{G}_v)$, and $\mathfrak{R}_N(\mathcal{H}_v)$ are trained within the same loss function class and are independent of $w_v$, for $0 < \delta < 1$, with probability at least $1 - \delta$, to ensure that the generalization bound $L_y$ under $w_v^d$ is smaller than that under $w_v^s$, it is required that:

$$\underbrace{\mathbb{E}[w_v^d] \stackrel{\text{\tiny $\doteq$}}{} w_v^s}_{\text{always holds}}, \quad \text{Cov}\left( w_v, \ell_y\left( G(\mathbf{c}^{(v)}), \mathbf{y} \right) \right) \leq 0,$$
$$(12)$$
$$\text{Cov}(w_v, \left\| \hat{\mathbf{c}}^{(v)} - \mathbf{c}^{(v)} \right\|_1) \leq 0.$$

Although $\ell_y \left( G(\mathbf{c}^{(v)}), \mathbf{y} \right)$ denotes the prediction loss

obtained from the true concepts, $\ell_y \left( G(\mathbf{c}^{(v)}), \mathbf{y} \right)$ and $\ell_y \left( G(\hat{\mathbf{c}}^{(v)}), \mathbf{y} \right)$ are positively correlated, since a smaller deviation between $\hat{\mathbf{c}}^{(v)}$ and $\mathbf{c}^{(v)}$ leads to closer prediction behavior of $G(\hat{\mathbf{c}}^{(v)})$ and $G(\mathbf{c}^{(v)})$, which in turn results in similar values of the two losses. In addition, the concept loss (L1loss) is required to be negatively correlated $w_v$. Thus, we present the following corollary:

**Corollary 1** When the fusion weight $w_v = w_v^d$ exhibits a negative correlation with both the prediction and concept losses, the generalization bound of multi-view fusion loss becomes lower than that of static fusion $w_v = w_v^s$.

Drawing upon the potential negative correlation between Dirichlet uncertainty and classification loss discussed in (Sensoy et al., 2018; Zhang et al., 2023), we quantify concept and grading uncertainties respectively based on Dirichlet theory. Together with Corollary 1, we propose the dual uncertainty-aware multi-view fusion module. The proof of Equ. (11) is provided in Appendix A.1.

### 3.3.2. DUAL UNCERTAINTY-AWARE MODULE

For each view, we quantify concept- and grading-level uncertainty under the evidential framework of Subjective Logic, which parameterizes belief masses via a Dirichlet distribution (Smith & Shafer, 1976). For concept-level modeling, we treat each concept as a binary classification. The evidence vector $\mathbf{e}_{v,c_j} = [e_{v,c_j}^+, e_{v,c_j}^-] = softplus(\hat{\mathbf{c}}^{(v)})$ yields Dirichlet parameters $\alpha_{v,c_j}^+ = e_{v,c_j}^+ + 1$ and $\alpha_{v,c_j}^- = e_{v,c_j}^- + 1$. The belief masses and uncertainty for concept $j$ in view $v$ are:

$$b_{v,c_j}^+ = \frac{\alpha_{v,c_j}^+ - 1}{S_{v,c_j}}, \quad b_{v,c_j}^- = \frac{\alpha_{v,c_j}^- - 1}{S_{v,c_j}}, \quad \psi_{v,c_j}^{\text{con}} = \frac{2}{S_{v,c_j}},$$
$$(13)$$

where $S_{v,c_j} = \alpha_{v,c_j}^+ + \alpha_{v,c_j}^-$. The overall concept uncertainty for view $v$ is averaged over all $m$ concepts: $\Psi_v^{\text{con}} = \frac{1}{m} \sum_{j=1}^{m} \psi_{v,c_j}^{\text{con}}$. For grading-level modeling with $K$ classes, the evidence vector $\mathbf{e}_v^{\text{gr}} = [e_v^{(1)}, \dots, e_v^{(K)}] = \text{softplus}(\hat{\mathbf{y}}^{(v)})$ gives $\alpha_v^{(i)} = e_v^{(i)} + 1$. The belief mass for grade $i$ and the grading uncertainty are:

$$b_v^{(i)} = \frac{\alpha_v^{(i)} - 1}{S_{v,gr}}, \quad \psi_v^{\text{gr}} = \frac{K}{S_{v,gr}}, \quad (14)$$

with total strength $S_{v,gr} = \sum_{i=1}^{K} \alpha_v^{(i)}$, satisfying $\sum_{i=1}^{K} b_v^{(i)} + \psi_v^{\text{gr}} = 1$.

### 3.3.3. MULTI-VIEW DECISION-MAKING UNDER DUAL UNCERTAINTIES

To construct a fully interpretable multi-view fundus decision model, our method exploits the uncertainties $\psi_v^{\text{con}}$ and $\psi_v^{\text{gr}}$ to assess view reliability, which in turn guides the dynamic

fusion of outputs across views. In particular, the final grading decision $\hat{y}$ is obtained by summing the view-specific outputs $\hat{y}^{(v)}$, each weighted by a reliability score that combines concept- and grading-level certainties, $(1 - \psi_v^{\text{con}})$ and $(1 - \psi_v^{\text{gr}})$, with a learnable parameter $W_c = \frac{e^{-\eta}}{1+e^{-\eta}} > 0$ controlling their trade-off:

$$\hat{y} = \sum_{i=1}^{V}[W_c(1 - \psi_v^{\text{con}}) + (1 - W_c)(1 - \psi_v^{\text{gr}})] \odot \hat{y}^v. \quad (15)$$

This inverse dual-uncertainty design offers two primary advantages: 1) it enhances diagnostic accuracy by reducing the weight of views associated with high predictive grading uncertainty; 2) it simultaneously diminishes the influence of views with low interpretability, as low-confidence lesion concepts would otherwise undermine the reliability of the interpretable reasoning pipeline $\hat{c}^{(v)} \rightarrow \hat{y}^{(v)}$.

### 3.4. Loss function

The training objective combines concept-level supervision for each view and the overall grading supervision. Specifically, we minimize $\mathcal{L} = \mathcal{L}_{FL_\gamma}(y, \hat{y}) + \alpha \mathcal{L}_{BCE}(c, \hat{c})$. Here, the first term corresponds to the *Focal Loss* (with focusing parameter $\gamma$) for class-imbalanced DR grading, and the second term corresponds to the *Binary Cross-Entropy Loss* for concept prediction, with $\alpha$ balancing the two. By jointly optimizing both terms, the model is encouraged to learn faithful concept representations while simultaneously improving the final grading performance. Detailed hyperparameter experiments are presented in Fig. 7.

### 3.5. Multi-View Test time Invention

Building upon our reasoning chain and dual-uncertainty decision paradigm, we propose a multi-view intervention mechanism that enables physicians to intervene on either a single view or a specific concept. Our method not only retains the ability of single-view CBM to intervene on concepts to influence single-view decisions, but also leverages dual uncertainties at both the concept and grading levels to increase the contribution of the corresponding view to the overall decision. Specifically, taking view $i$ as an example, if an ophthalmologist corrects the result $\hat{c}^{(v)}$ of lesion concepts in this view, the DR grading can first be re-inferred and updated as $\hat{y}^{(v)}$, after which the dual uncertainties of the view are updated accordingly, thereby influencing the fused diagnostic outcome $\hat{y}'$.

## 4. Experiments

### 4.1. Experimental Settings

**Datasets.** We evaluate our method on the two publicly available multi-view DR grading datasets, MFIDDR (Luo

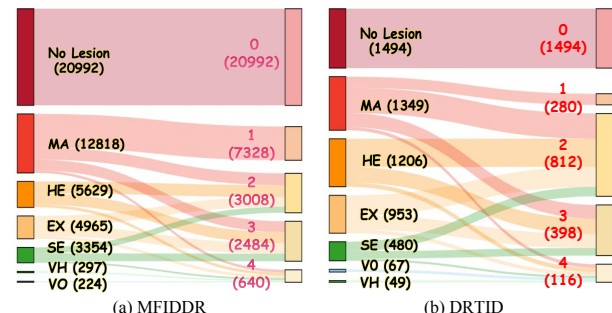

(a) MFIDDR      (b) DRTID

*Figure 3.* The data correlation and distribution of lesion concepts and DR grades in the two benchmarks. On the left are lesion concepts, and on the right are DR grades. The text indicates the class name, and the number in parentheses denotes the number of samples in each class.

et al., 2023) and DRTiD (Hou et al., 2022). MFIDDR contains 34,452 images from 4,344 patients, annotated with five DR grades across four standard views (macula-centered, optic disc–centered, and superior/inferior tangent to the optic disc). DRTiD consists of 3,100 paired macula- and optic disc–centered images from 1,550 eyes. To enable concept-based reasoning, ophthalmologists annotate six lesion concepts in the fundus images of each dataset, which serve as concept prediction labels: EX, HE, MA, SE, VH, and VO. For fair comparison, we follow the official data split protocols provided by each dataset. Detailed statistics of the dataset distributions are summarized in Fig. 3 and Appendix A.2. And lesion concept masks are generated by the HACDR-Net (Xu et al., 2024) pre-trained on DDR (Li et al., 2019) dataset.

**Evaluation Metric and Compared Methods.** In this section, we evaluate multi-view DR diagnosis on two tasks: multi-view DR grading and lesion concept classification. Grading is assessed using accuracy (Acc), specificity (Spe), kappa, macro-F1, and AUC (Trevethan, 2017), and the lesion concept classification uses AUPR, AUC, Acc, macro-F1, and ranking loss (RL). Furthermore, we report on the inference time in the MFIDDR dataset. To facilitate a comprehensive evaluation, the comparison methods are divided into two categories: (i) non-interpretable multi-view DR diagnosis methods, including CrossFit (Hou et al., 2022), ETMC (Han et al., 2022), MVCINN (Luo et al., 2023), Retfound (Zhou et al., 2023), CVSRA-ViT (Lin et al., 2025), SMVDR (Luo et al., 2025), and WMIMVDR (Hu et al., 2025), and (ii) interpretable multi-view DR diagnosis methods, including Multi-Task (Liu et al., 2019), MVCBM (Klimiene et al., 2022a), CEM (Espinosa Zarlenga et al., 2022), PCBM-h (Yuksekgonul et al., 2023), SSMVCBM (Marcinkevičs et al., 2024), and CLAT (Wen et al., 2024).

**Implementation Details.** All experiments are implemented with PyTorch and conducted on a NVIDIA RTX 4090 GPU.

*Table 1.* Comparison of Accuracy, Specificity, Kappa, and F1-score on MFIDDR and DRTiD, and inference time for different models on DR grading.

| Method | Venue | Backbone | MFIDDR (four views) | | | | DRTiD (two views) | | | | Infer. Time ↓ |
|---|---|---|---|---|---|---|---|---|---|---|---|
| | | | Acc↑ | Spe↑ | Kappa↑ | F1↑ | Acc↑ | Spe↑ | Kappa↑ | F1↑ | |
| Non-interpretable Multi-View DR Diagnosis Methods | | | | | | | | | | | |
| CrossFit | BIBM'22 | Resnet-50 | – | – | – | – | 72.73 | 86.63 | 57.60 | 70.53 | – |
| ETMC | TPAMI'22 | Resnet-50 | 81.54 | 83.44 | 64.76 | 79.74 | 65.48 | 78.14 | 44.79 | 61.35 | **6.61** |
| MVCINN | AAAI'23 | Resnet-50+ViT | 80.10 | 83.32 | 62.45 | 78.86 | 68.18 | 85.78 | 51.39 | 66.83 | 31.31 |
| CVSRA-ViT | PR'25 | VGG+ViT | 82.61 | 86.77 | 68.57 | 81.94 | 70.62 | 88.91 | 55.74 | 69.97 | 71.53 |
| SMVDR | AAAI'25 | Mamba | 84.01 | 91.30 | 71.36 | 83.69 | 74.52 | 92.29 | 61.38 | 72.86 | 65.71 |
| WMIMVDR | ICME'25 | Resnet-50+ViT | 84.15 | 89.95 | 71.16 | 83.59 | 73.23 | 90.58 | 58.87 | 70.62 | 25.44 |
| Interpretable Multi-View DR Diagnosis Methods | | | | | | | | | | | |
| Multi-Task | CVPR'19 | Resnet-50 | 83.73 | 89.06 | 70.24 | 83.12 | 72.79 | 89.32 | 56.98 | 70.12 | 8.24 |
| MVCBM | ICML'22 | Resnet-50 | 83.22 | 88.22 | 69.12 | 82.43 | 71.54 | 85.02 | 57.89 | 68.50 | 19.67 |
| CEM (MV) | NeurIPS'22 | Resnet-50 | 84.12 | 88.77 | 70.83 | 83.45 | 74.55 | 91.67 | 61.42 | 72.06 | 21.25 |
| PCBM (MV) | ICML'23 | Resnet-50 | 83.52 | 91.19 | 70.35 | 83.29 | 74.73 | 90.26 | 60.68 | 71.99 | 17.56 |
| SSMVCBM | MIA'24 | Resnet-50 | 82.75 | 85.81 | 67.55 | 81.51 | 73.98 | 91.74 | 60.01 | 70.81 | 20.71 |
| CLAT (MV) | TMI'25 | ViT | 82.89 | 86.66 | 68.16 | 81.88 | 74.55 | 91.33 | 61.03 | 72.77 | 33.02 |
| **ProConMV** | **ICML'26** | **Hilbert-RWKV** | **86.75** | **92.79** | **76.05** | **86.35** | **76.77** | **93.77** | **64.47** | **74.64** | 8.77 |

*Table 2.* Comparison of AUPR, Accuracy, F1-score, and Ranking Loss on MFIDDR and DRTiD for different models on lesion concept classification. The best results are highlighted in bold, and '(MV)' means transforming into a multi-view method. (Unit: %)

| Method | MFIDDR | | | | DRTiD | | | |
|---|---|---|---|---|---|---|---|---|
| | AUPR↑ | Acc↑ | F1↑ | RL↓ | AUPR↑ | Acc↑ | F1↑ | RL↓ |
| Multi-Task | 54.69 | 93.87 | 51.69 | 3.73 | 47.32 | 87.31 | 43.90 | 7.88 |
| MVCBM | 61.56 | 94.22 | 59.10 | 3.21 | 48.50 | 88.82 | 41.88 | 5.56 |
| CEM (MV) | 65.47 | 94.74 | 60.42 | 2.86 | 48.50 | 89.95 | 44.63 | 6.12 |
| PCBM (MV) | 68.12 | 94.85 | 66.08 | 1.91 | 52.59 | 90.46 | 47.24 | 4.52 |
| SSMVCBM | 66.25 | 94.42 | 63.34 | 2.17 | 53.52 | 90.35 | 47.15 | 4.25 |
| CLAT (MV) | 63.89 | 94.63 | 59.15 | 2.98 | 51.83 | 89.97 | 46.82 | 4.71 |
| **ProConMV** | **72.26** | **95.42** | **68.43** | **1.45** | **55.86** | **90.83** | **48.00** | **3.42** |

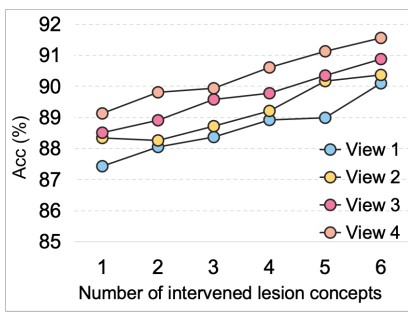

*Figure 4.* Performance evaluation of the multi-view test-time intervention. Points are colored by the number of intervened views.

We resize the images to the resolution of $256 \times 256$. The batch size and number of epochs are set to 8 and 100, respectively. The Adam optimizer is used with an initial learning rate of $10^{-5}$, which is adjusted by a cosine annealing scheduler. We select the model achieving the best grading performance on the validation set as the final model, which is then used for subsequent testing and analysis.

### 4.2. Experimental Analysis

**Comparison with Advanced Methods.** We compare our method with twelve state-of-the-art multi-view methods on two datasets. As shown in Table 1, our ProConMV achieves the best performance in multi-view DR grading on two datasets. Specifically, ProConMV improves accuracy by 2.6% on the four-view dataset and by 2.04% on DRTiD, with the highest Kappa improvement of 4.69% on MFIDDR. As presented in Table 2, our method also achieves the best results in lesion concept classification on both datasets. The AUPR is improved by 4.14% and 2.34% on MFIDDR and DRTiD, respectively, while RL and HL

loss are significantly reduced. Benefiting from the linear complexity of Hilbert-RWKV, our method also ranks among the sub-optimal in inference time. Futhermore, the interpretable inference process visualization of ProConMV is provided in Fig. 6. Overall, our method achieves the highest accuracy in both grading and concept prediction tasks, and also delivers superior inference efficiency compared to existing interpretable and non-interpretable multi-view methods.

**Analysis of Test-time Intervention Capability.** As a concept-based model, our approach allows for interventions at the level of per-view concepts. Specifically, ophthalmologists can revise the diagnosis of a particular view, a specific lesion type, or the same lesion type across multiple views, and such interventions directly refine the final DR grade. As shown in Fig. 4, the overall grading accuracy increases proportionally with the number of intervened views and lesion concepts. When all erroneous concepts are corrected, the grading accuracy reaches 91.56%, yielding a 4.81% improvement over the non-intervention setting. This verifies the effectiveness of the proposed intervention method.

*Table 3.* Ablation studies of key modules on MFIDDR. × and ✓ denote the absence and presence of each module. The first row corresponds to the baseline, MVCBM. (Unit: %)

| Mask | Text | Hilbert-RWKV | VT-RWKV | DU-MVFD | Grading | | Concept Pred. | |
|---|---|---|---|---|---|---|---|---|
| | | | | | Acc | Kappa | AUPR | F1 |
| ✓ | × | × | × | × | 83.22 | 69.12 | 61.56 | 59.10 |
| ✓ | × | ✓ | × | × | 85.72 | 74.31 | 62.90 | 63.87 |
| ✓ | ✓ | × | ✓ | × | 85.45 | 74.29 | 64.72 | 62.61 |
| ✓ | × | × | × | ✓ | 84.15 | 71.57 | 60.87 | 58.92 |
| ✓ | ✓ | ✓ | ✓ | × | 86.23 | 75.23 | 70.69 | 67.89 |
| ✓ | × | ✓ | × | ✓ | 85.68 | 74.43 | 70.12 | 64.39 |
| ✓ | ✓ | × | ✓ | ✓ | 85.72 | 72.68 | 67.71 | 66.12 |
| × | ✓ | ✓ | ✓ | ✓ | 85.21 | 73.98 | 68.59 | 66.26 |
| ✓ | ✓ | ✓ | ✓ | ✓ | **86.75** | **76.05** | **72.26** | **68.43** |

*Table 4.* Comparison of backbones on MFIDDR. (Unit: %)

| Backbone | Grading | | Concept Pred. | | Params | Infer. |
|---|---|---|---|---|---|---|
| | Acc | F1 | AUPR | F1 | (M) | (ms) |
| VGG-16 | 85.63 | 85.26 | 67.69 | 66.88 | 15.29 | **6.93** |
| ResNet-50 | 85.72 | 84.08 | 67.71 | 66.12 | 25.26 | 10.16 |
| ViT-B | 85.82 | 85.44 | 55.01 | 52.81 | 86.61 | 12.09 |
| Swin v1-S | 86.09 | 86.03 | 66.65 | 60.09 | 49.56 | 18.65 |
| Swin v2-S | 86.33 | 85.76 | 61.47 | 57.52 | 37.93 | 26.95 |
| VMamba | 83.03 | 82.29 | 59.43 | 52.15 | 14.60 | 9.27 |
| **Hilbert-RWKV** | **86.75** | **86.35** | **72.26** | **68.92** | **6.70** | 8.77 |

## 4.3. Ablation Study

**Ablation Results of Hilbert-RWKV.** We first compare Hilbert-RWKV with existing state-of-the-art backbones, as shown in Table 3 and 4. Compared to ResNet-50 (He et al., 2016), ViT-Big (Dosovitskiy et al., 2021), VMamba-Tiny (Liu et al., 2024), Swin v1-S (Liu et al., 2021), and Swin v2-S (Liu et al., 2022b), our method achieves consistent improvements in DR grading and lesion classification with lower parameter counts and shorter inference times. Hilbert-RWKV demonstrates a superior capacity for detecting tiny lesions, outperforming the runner-up (ResNet-50) by 4.55% in AUPR and 2.80% in F1-score during concept classification. Notably, these performance gains are accompanied by a significantly smaller parameter footprint and lower computational overhead (6.70M vs 25.26M, 8.77ms vs 10.16ms). Compared with Hilbert-curve, sweep, zigzag (Chen et al., 2025), and unidirectional scanning lead to drops of 0.78% and 2.18% in F1 scores for DR grading and lesion classification, respectively, as presented in Table 5. These results substantiate the superiority of the Hilbert-RWKV design in the capture of lesion concepts, diagnostic reasoning, and computational efficiency.

**Ablation Results of VT-RWKV and DU-MVFD.** As illustrated in Fig. 5(a), we evaluate various interaction strategies for VT-RWKV. Compared to channel-wise fusion (Cat), cross-attention (Attn (Wei et al., 2020)), and dynamic multimodal fusion (DyF (Xue & Marculescu, 2023)), our pro-

*Table 5.* Comparison of RWKV scanning strategies on the MFIDDR dataset. (Unit: %)

| Strategy | Grading | | Concept Pred. | |
|---|---|---|---|---|
| | Acc | F1 | F1 | AUPR |
| Sweep | 86.29 | 85.94 | 65.24 | 69.61 |
| Horizontal | 86.24 | 86.05 | 64.93 | 70.97 |
| Vertical | 86.10 | 85.96 | 64.75 | 70.46 |
| Zigzag | 86.24 | 85.70 | 63.89 | 67.60 |
| **Hilbert** | **86.75** | **86.35** | **68.92** | **72.26** |

posed VT-RWKV consistently yields superior results, with marked improvements of 0.70% Acc, 1.13% Kappa, 0.72% AUPR, and +1.61% F1. These results validate the efficacy of our image-text concept interaction mechanism. Regarding multi-view fusion (Fig. 5(b)), the dual-uncertainty module attains peak performance, outstripping the Baseline (Concat), Late Fusion, TMC, and MoE (Cao et al., 2023) by 0.86% Acc, 1.97% Kappa, 2.57% AUPR, and 1.77% F1. This experiment underscores the critical role of uncertainty modeling in enhancing robust multi-view decision fusion.

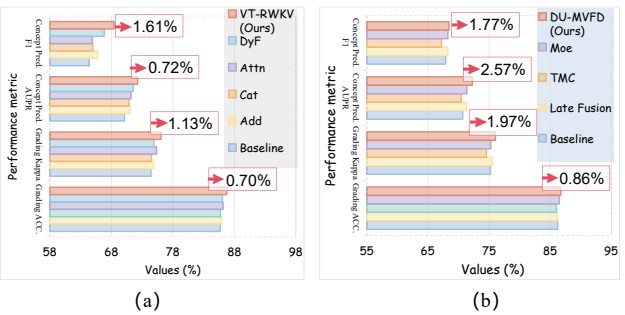

*Figure 5.* Results of the ablation study on (a) VT-RWKV and (b) DU-MVFD on the MFIDDR dataset.

## 5. Limitation

Currently, our model still relies on lesion concept annotations for supervised training, which may increase annotation costs and limit its applicability to scenarios without concept labels. In future work, we will explore non-supervised concept learning to reduce the dependence on full labels.

## 6. Conclusion

This paper proposes a full-process interpretable model, ProConMV. It achieves deep extraction and fusion of multi-source features, introduces lesion concepts to construct a causal reasoning chain, and incorporates real-time physician intervention. Moreover, the proposed multi-view decision-making approach theoretically reduces generalization error and achieves traceability through a dual uncertainty module. The results show that it achieves state-of-the-art performance and high clinical credibility.

## Acknowledgments

This work was supported by the National Natural Science Foundation of China under Grant No. 62502320, the Natural Science Foundation of Guangdong Province under Grant No. 2025A1515010184, the project of Shenzhen Science and Technology Innovation Committee under Grant No. JCYJ20240813141424032, and the Scientific Foundation for Youth Scholars of Shenzhen University Grant No. 827-0001083.

## Impact Statement

This paper aims to advance interpretable and provenance-enabled AI for multi-view diabetic retinopathy diagnosis. Our contributions enhance model transparency and reliability in safety-critical healthcare applications by integrating concept-based reasoning, uncertainty estimation, and physician intervention. This work promotes responsible AI deployment by enabling clinicians to understand, verify, and trust diagnostic decisions, especially in retinal screening scenarios where interpretability and reliability are crucial.

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

# A. Appendix

### A.1. Proof of Theorem 1

According to the definitions and settings mentioned above, based on the convexity of the prediction loss $\ell_y^{(v)}(\cdot, \cdot)$ and the normalization property of $w_v$, we can derive:

$$\ell_y\left(\sum_{v=1}^{V} w_v\, G(\hat{\mathbf{c}}^{(v)}),\, \mathbf{y}\right) \leq \sum_{v=1}^{V} w_v \ell_y(G(\hat{\mathbf{c}}^{(v)}),\, \mathbf{y}) \tag{16}$$

Using the Lipschitz constraint, we decompose $\ell_y\big(G(\hat{\mathbf{c}}^{(v)}), \mathbf{y}\big)$ as follows:

$$\ell_y\big(G(\hat{\mathbf{c}}^{(v)}), \mathbf{y}\big) \;\leq\; \ell_y\big(G(\mathbf{c}^{(v)}), \mathbf{y}\big) + L_g^{(v)}\big\|\hat{\mathbf{c}}^{(v)} - \mathbf{c}^{(v)}\big\|_1 \;\leq\; \ell_y\big(G(\mathbf{c}^{(v)}), \mathbf{y}\big) + L_g^{(v)}\,\ell_c^{(v)}. \tag{17}$$

By combining Equ. (16) and (17), the upper bound of $L_y$ can be rewritten as:

$$L_y \leq \sum_{v=1}^{V} \mathbb{E}\Big[\, w_v\, \ell_y(G(\hat{\mathbf{c}}^{(v)}), y)\Big] \leq \sum_{v} \mathbb{E}\Big[w_v\, \ell_y(G(\mathbf{c}^{(v)}), \mathbf{y})\Big] \;+\; \sum_{v=1}^{V} L_g^{(v)}\, \mathbb{E}\Big[w_v\, \ell_c^{(v)}\Big]. \tag{18}$$

According to the property of expectation, for any random variables $A$ and $B$, $\mathbb{E}[AB] = \mathbb{E}[A]\,\mathbb{E}[B] + \mathrm{Cov}(A, B)$.

$$L_y \leq \sum_{v=1}^{V} \big(\mathbb{E}[w_v]\,\mathbb{E}[\ell_y\big(G(\mathbf{c}^{(v)}), \mathbf{y}\big)]\big) + \sum_{v=1}^{V} \big(\mathbb{E}[w_v]\, L_g^{(v)}\, \mathbb{E}[\ell_c^{(v)}]\big)$$

$$+ \sum_{v=1}^{V} \mathrm{Cov}(w_v, \ell_y\big(G(\mathbf{c}^{(v)}), \mathbf{y}\big)) + \sum_{v=1}^{V} L_g^{(v)}\, \mathrm{Cov}(w_v, \ell_c^{(v)}). \tag{19}$$

To simplify Equ. (19), we take $\mathbb{E}[\ell_y\big(G(\mathbf{c}^{(v)}), \mathbf{y}\big)]$ as an example and invoke Rademacher complexity theory, which establishes that with a confidence level $1 - \delta$, where $0 < \delta < 1$, the following holds:

$$\mathbb{E}[\ell_y\big(G(\mathbf{c}^{(v)}), \mathbf{y}\big)] \;\leq\; \hat{L}_y^{(v)} \;+\; \mathfrak{R}_N(\mathcal{G}) \;+\; P\sqrt{\frac{\ln(V/\delta)}{N}}. \tag{20}$$

Where $\hat{L}_y^{(v)}$ denotes the empirical prediction error under correct concepts. Similarly, it can be derived that: $\mathbb{E}[\ell_c^v] \;\leq\; \hat{L}_c^{(v)} \;+\; \mathfrak{R}_N(\mathcal{H}_v) \;+\; P\sqrt{\frac{\ln(V/\delta)}{N}}$. In summary, we can obtain the final theorem:

$$L_y \leq \underbrace{\sum_{v=1}^{V} \mathbb{E}[w_v]\hat{L}_y^{(v)} + \sum_{v=1}^{V} \mathbb{E}[w_v]L_g^{(v)}\hat{L}_c^{(v)}}_{\text{Term-L (average empirical loss of prediction and concept)}} \;+\; \underbrace{\sum_{v=1}^{V} \mathbb{E}[w_v]\mathfrak{R}_N(\mathcal{G}) + \sum_{v=1}^{V} \mathbb{E}[w_v]L_g^{(v)}\mathfrak{R}_N(\mathcal{C})}_{\text{Term-C (average complexity of prediction and concept)}}$$

$$+ \underbrace{\sum_{v=1}^{V} \mathrm{Cov}(w_v, \ell_y\big(G(\mathbf{c}^{(v)}), \mathbf{y}\big)) + \sum_{v=1}^{V} L_g^{(v)}\, \mathrm{Cov}(w_v, \ell_c^{(v)})}_{\text{Term-Cov (covariance between fusion weights and losses)}} + \underbrace{2P\sqrt{\frac{\ln(V/\delta)}{N}}}_{\text{concentration term}}. \tag{21}$$

Consequently, we derive the upper bound of $L_y$ in terms of Rademacher complexity.

### A.2. Statistical information of two datasets

In Fig. 3, we list the detailed information of two DR diagnosis datasets used in our experiments. The relationships represented by the Sankey diagram capture underlying DR diagnostic rules, which in turn validate the rationality and interpretability of our reasoning model. The Sankey diagrams illustrate the diagnostic rules by mapping lesion concepts to DR grades across the MFIDDR and DRTID benchmarks. The flow distributions reveal that the absence of lesions is predominantly associated with Grade 0, while MA serves as the primary indicator for early-stage DR, specifically Grades 1 and 2. More advanced lesions, such as HE, EX, and SE, show a clear progression toward moderate and severe stages, corresponding to Grades 2 and 3. Finally, high-risk manifestations like VH and VO lead directly to Grade 4 classifications, demonstrating that the model's internal reasoning consistently aligns with established clinical guidelines.

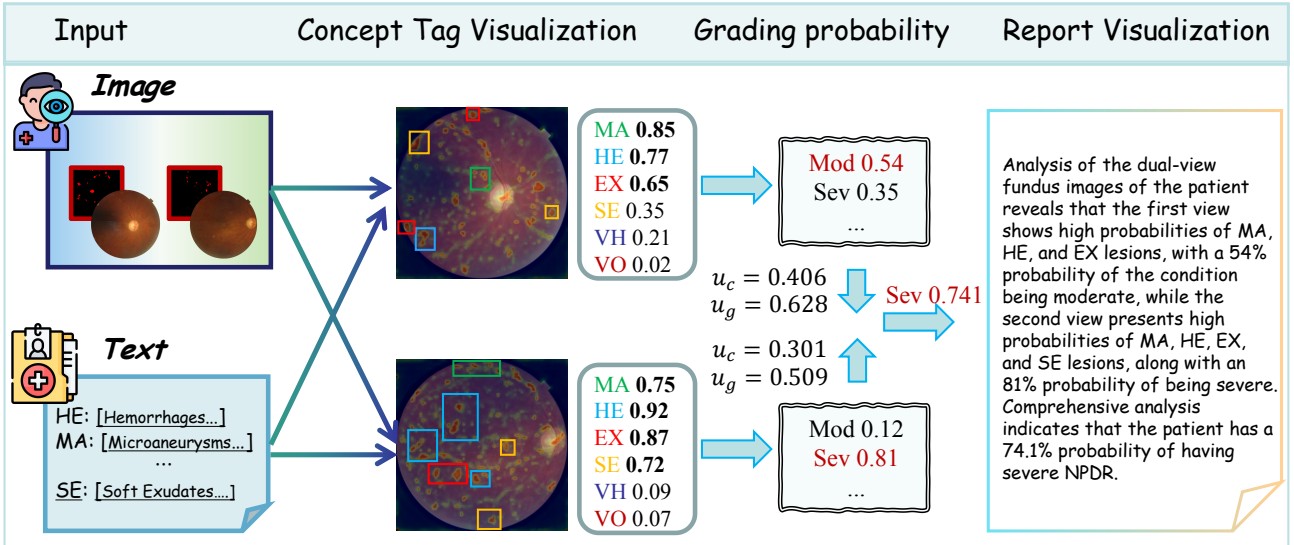

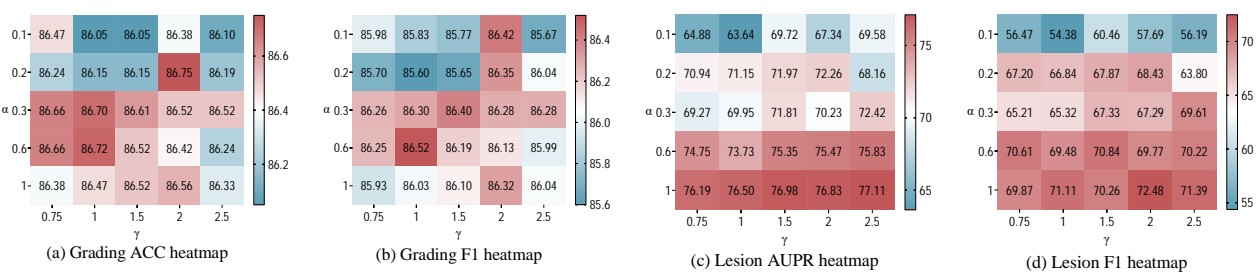

Figure 6. Visualization display of our ProConMV's inference process.

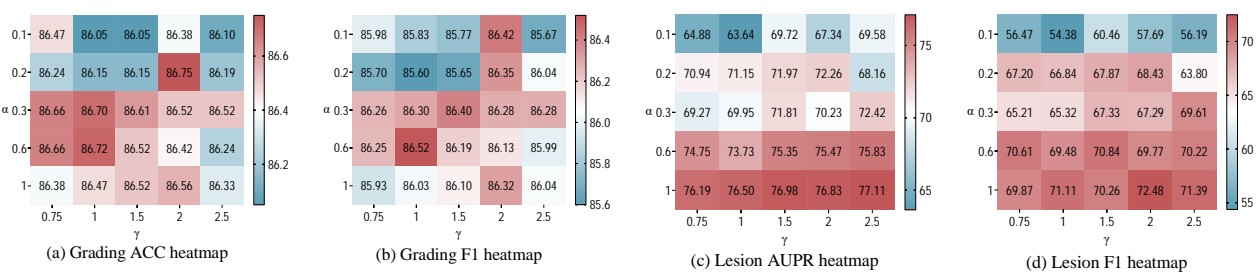

(a) Grading ACC heatmap  (b) Grading F1 heatmap  (c) Lesion AUPR heatmap  (d) Lesion F1 heatmap

Figure 7. Results of the ablation study on the hyperparameters $\alpha$ and $\gamma$. The figure presents grading accuracy, grading F1-score, concept accuracy, and concept F1-score under different values of $\alpha$ and $\gamma$. The best results are indicated by the warmest colors.

## A.3. Interpretability Analysis

Fig. 6 illustrates that our model provides a fine-grained and clinically meaningful interpretation of its diagnostic workflow. For each fundus view, the model first localizes lesion regions and assigns concept tags (e.g., MA, HE, EX, SE, VH, VO) with calibrated probabilities, which are consistent with the accompanying textual descriptions of hemorrhages, microaneurysms, and soft exudates. In the first view, high probabilities of MA, HE, and EX lead to a moderate grading score, whereas in the second view, simultaneously elevated MA, HE, EX, and SE scores support a severe grade. The dual-uncertainty-aware fusion module then combines per-view concept and grading confidences to produce a final severe NPDR probability of 0.741, explicitly revealing how each view contributes to the outcome. Finally, the system generates a natural-language report that summarizes the detected lesions, per-view severity, and overall diagnostic conclusion, allowing clinicians to verify the prediction step by step. Together, these visual and textual explanations show that the model's decision is driven by clinically interpretable lesion patterns rather than opaque global features.

## A.4. Hyperparameter Analysis

The hyperparameters $\alpha$ and $\gamma$ respectively control the weight of the concept prediction loss and the focusing parameter in the grading loss. To analyse the impact of these hyperparameters, we conduct a two-dimensional ablation study by systematically varying both $\alpha$ and $\gamma$. As shown in Fig. 7, when $\alpha$ increases, the contribution of the concept prediction loss becomes more prominent, resulting in higher concept prediction accuracy. However, a larger value of $\alpha$ reduces grading accuracy, which contradicts our goal of achieving optimal grading performance. Therefore, there is a trade-off between these two hyperparameters. Based on this analysis, we select $\alpha = 0.2$ and $\gamma = 2.0$ as the optimal hyperparameter settings.

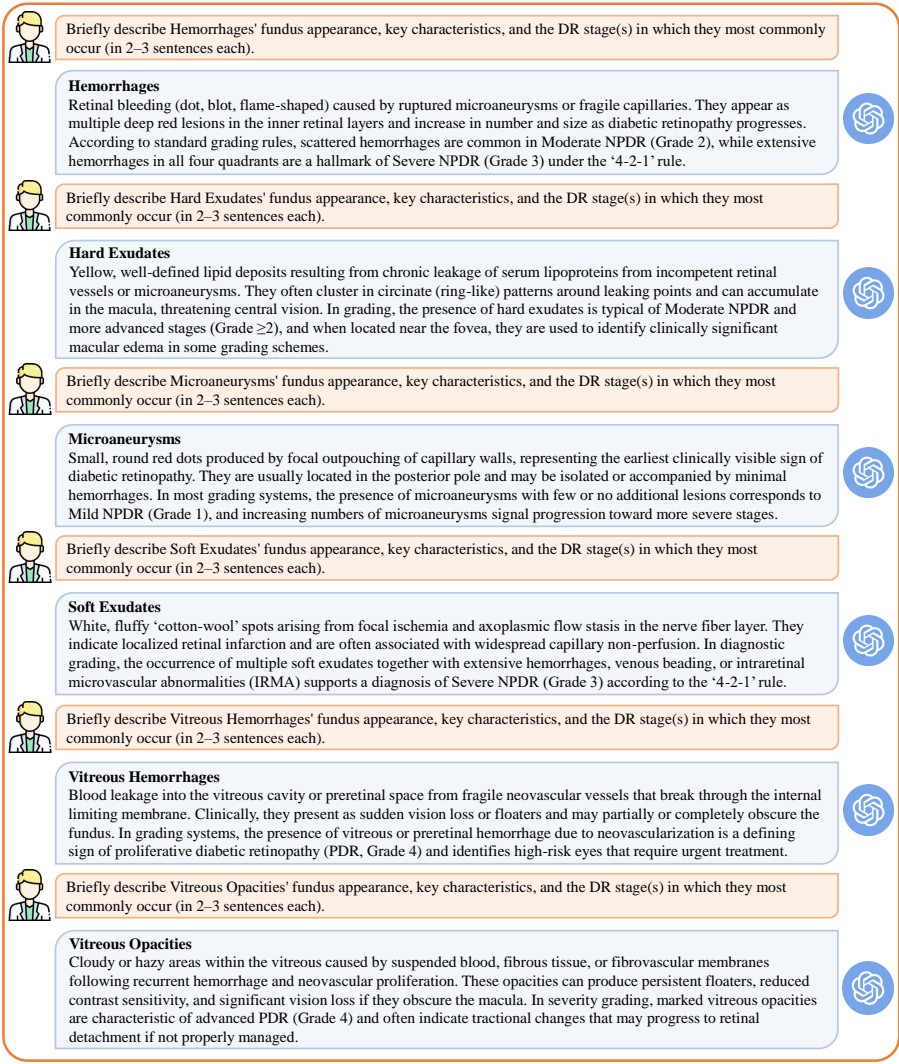

*Figure 8.* The generation of concept text.

## A.5. Concept Text Generation

To accurately and systematically capture the medical characteristics of DR lesions, we conducted a structured six-round interactive dialogue with GPT-5 (Singh et al., 2025). This approach addresses the shortcomings of manual feature extraction, avoiding subjective bias, incomplete information coverage and inconsistent standards. The dialogue follows a logic of hierarchical refinement. We first categorize core DR lesion types through macro queries, such as hemorrhages, exudates, microaneurysms and intraretinal microvascular abnormalities. Each subsequent round of dialogue delves into more detailed information, covering the lesion's typical anatomical location, morphological features, color properties, size range and key points for distinguishing it from confusing structures. We explicitly guided GPT-5 to follow clinical standards, such as the International Clinical Diabetic Retinopathy Severity Scale (ICDR), and integrate relevant evidence from authoritative ophthalmic guidelines to ensure the generated descriptions are medically rigorous. These descriptions include not only the baseline appearance and key diagnostic features of lesions but also their dynamic manifestations across different DR stages. For example, microaneurysms develop from small, discrete dots into clustered lesions, and hard exudates transform into lipid-laden plaques in advanced stages of the disease. As shown in Fig. 8, the hierarchical structure and core content of the generated text can provide a reliable semantic foundation for downstream tasks.

