# OpenReview forum: "ProConMV: Provenance-Enabled Conceptual Framework for Interpretable Multi-View Diabetic Retinopathy Diagnosis"
_ICML.cc/2026/Conference — ICML 2026 regular_

### Official Review · Reviewer_oRht · 2026-03-10

**Soundness:** 4
**Presentation:** 4
**Significance:** 3
**Originality:** 3
**Overall Recommendation:** 6
**Confidence:** 5

**Summary:**

This work proposes ProConMV, a provenance-enabled framework for interpretable multi-view diabetic retinopathy (DR) diagnosis. By integrating fundus images, lesion masks, and structured clinical text, ProConMV establishes a semantically rich foundation. The framework leverages a Hilbert RWKV backbone to capture fine-grained lesion features with linear complexity, and aligns visual perception with expert medical knowledge through a visual-text concept interaction module. By quantifying concepts and grading uncertainties based on generalization theory, ProConMV achieves reliable dynamic multi-view fusion, supporting real-time doctor intervention to ensure traceable and verifiable decisions.

**Compliance With Llm Reviewing Policy:**

Affirmed.

**Ethics Expertise Needed:**

["Other Expertise"]

**Final Justification:**

The authors have provided targeted and effective revisions according to the comments. They have enriched the algorithm descriptions with more standard mathematical expressions, and optimized the discussion on parameter ablation studies. The overall logical structure of the manuscript is more compact, the experimental results are presented more intuitively, and the technical content is more comprehensive and convincing. For these reasons, I suggest raising the evaluation score appropriately.

**Key Questions For Authors:**

1)	The concept sets in this work is static. Why not design dynamic concept sets?

2)	What motivates the incorporation of the Hilbert curve into the RWKV architecture?

3)	I find that many concept-based models employ CLIP for concept generation. You should delineate the key distinctions between these existing approaches and our proposed method.

**Limitations:**

See the weakness.

**Strengths And Weaknesses:**

Strengths:
1)	The core proposition is clearly explained in this paper, and the motivations behind it seem reasonable. Considering the emphasis on interpretability, the integration of text-prompting and interpretable multi-view fusion strategies is a well-justified and effective approach.
2)	The overall framework is built upon the Concept Bottleneck Model (CBM) and extends its application to the multi-view domain. And the overall quality of the illustrations and writing is high.
3)	The highlight of this interpretable framework is the incorporation of generalization theory. Guided by this theory, authors propose a design where concept-decision dual-uncertainty is inversely proportional to the class loss. By integrating theoretical derivation with experimental results, the authors provide a solid justification for their fusion strategy. I have carefully reviewed the theoretical derivation and experiments of this section.

Weaknesses:
1)	Partial module designs and experiments lack sufficient detail, such as the generation of static text prompts and specific implementation details for reproducing the concept-based models.
2)	There is no visualization in the main text. Its interpretability is not intuitively demonstrated.
3)	The authors would better supplement experiments with other multi-view medical images.
4)	In Section 2.1.2, there is an issue with the Bi-WKV formula: the first term of the molecule contains $v$ (without an index), while the term itself contains $v_t$; the second exponent uses $k_i$, but for the current label, the index should be $k_t$. It appears to be a typo, clarification is needed.

---

> ### Author Rebuttal · Authors · 2026-03-31
>
> Thank you for your positive evaluation and valuable suggestions. In response to your concern, we provide the following clarification.
>
> W1: The specific problem setting and generation procedure of our textual prompts have already been described in Fig. 10 of the appendix. We will further incorporate the detailed encoding process into the main paper in the form of pseudocode. Regarding the reproduction details of the concept-based models, all methods use a ResNet-50 backbone, with multi-view fusion performed by summing the decision outputs. All other experimental settings, including the loss function, hyperparameters, batch size, optimizer, and training epochs, are kept identical for fairness. Like our method, these models are supervised by both concept labels and grading labels.
>
> W2: Due to space limitations, we included only one visualization in the appendix. In the revised version, we will add more visualizations to the appendix, including the reasoning process, intervention examples, and feature Grad-CAM results.
>
> W3: We appreciate your concern about the lack of evaluation on external datasets. Nevertheless, the proposed framework is general and can be applied to other multi-view medical scenarios, including skin and chest disease diagnosis. To better support this point, we will provide supplementary experiments on these tasks in the appendix of the revised paper.
>
> W4: We will carefully check the manuscript and correct this error.
>
> Q1: Existing dynamic concept set construction methods [1,2] largely depend on powerful general-purpose visual foundations, such as CLIP or LLM-driven systems. However, in medical imaging, fine-grained concepts (e.g., hard exudates) are difficult to capture accurately, even for medically pretrained CLIP models. Consequently, the practical performance of current dynamic concept set methods in medical domains remains limited.
>
> Q2: In spatial feature maps, the motivation for introducing multi-directional RWKV is closely related to the locality-preserving advantages of space-filling curves such as the Hilbert curve. Some studies have shown that the Hilbert curve more effectively maps multidimensional data to one dimension while minimizing the likelihood that spatially adjacent points are mapped far apart in the serialized sequence. For example, prior work by Moon et al. [3] and Haverkort & van Walderveen [4] rigorously analyzed the clustering and locality-preserving behavior of different curves, showing that Hilbert consistently achieves lower dilation and clustering number than Morton (Z-order) or raster scans. Inspired by these findings, we argue that a single raster-like RWKV pass over a 2D feature map imposes a strong directional bias, so that vertically or diagonally neighboring pixels can become distant in the 1D sequence, effectively increasing dilation and degrading local clustering. By instead performing RWKV along multiple directions (e.g., left-to-right, right-to-left, top-to-bottom, bottom-to-top) and aggregating the resulting states, we emulate the locality-preserving spirit of Hilbert-like mappings: for a fixed receptive “window,” most neighboring pixels remain close in at least one traversal. This multi-directional design yields more isotropic modeling of local fundus structures, which is beneficial for downstream network modeling and local feature aggregation.
>
> In addition, we conduct comprehensive quantitative experiments in Table 1 comparing Hilbert-RWKV with its lightweight substitutes and various scanning strategies. The results show that Hilbert-RWKV achieves the best performance on the multi-view fundus imaging task across all metrics, further validating the advantage of its strong global modeling with locality-preserving design.
>
> **Table 1. Comparison of scanning strategies on MFIDDR. (%)**
>
> |Strategy |Grading Acc.|Grading F1| Concept F1 | Concept AUPR |
> |---|---|---|---|---|
> |Sweep|86.29|85.94|65.24|69.61|
> |Horizontal|86.24|86.05|64.93|70.97|
> |Vertical|86.10|85.96|64.75|70.46|
> |Zigzag|86.24| 85.70 | 63.89 | 67.60 |
> |**Ours** | **86.75** | **86.35** | **68.92** | **72.26** |
>
> Q3: Although many existing concept-based models rely on CLIP or Siglip image–text similarity to construct dynamic concept sets, such approaches are less suitable for medical imaging because fine-grained lesion concepts are often difficult to capture accurately, even with medically CLIP or Siglip variants [5]. In contrast, our method uses clinician-aligned lesion concepts within a multi-view concept bottleneck framework. The explicit concept-to-diagnosis reasoning and dual-uncertainty modeling make it more appropriate for an interpretable medical diagnosis.
>
> [1] Hybrid Concept Bottleneck Models, CVPR 2025
>
> [2] Label-free concept bottleneck models, ICLR 2023
>
> [3] Analysis of the clustering properties of the Hilbert space-filling curve, TKDE 2001.
>
> [4] Locality-preserving properties of space-filling curves, Computational Geometry.
>
> [5] Sigmoid Loss for Language Image Pre-Training, CVPR 2023

---

> > ### Author Rebuttal · Reviewer_oRht · 2026-04-01
> >
> > All of my concerns have been addressed, and I will raise my score.

---

> > > ### Author Response · Authors · 2026-04-07
> > >
> > > Thank you for your positive feedback. We sincerely appreciate your time and thoughtful evaluation throughout the review process. We are very glad that our rebuttal has addressed all of your concerns, and we are grateful for your supportive assessment of our work.

---

### Official Review · Reviewer_YJf7 · 2026-03-11

**Soundness:** 2
**Presentation:** 2
**Significance:** 3
**Originality:** 3
**Overall Recommendation:** 4
**Confidence:** 3

**Summary:**

This paper proposes ProConMV, an interpretable framework for multi-view diabetic retinopathy (DR) diagnosis that simultaneously addresses three key challenges: single-source input, untraceable reasoning processes, and lack of result verification mechanisms. Methodologically, the authors integrate multi-view fundus images, lesion segmentation masks, and textual lesion descriptions. A Hilbert-RWKV-based visual backbone extracts fine-grained lesion representations, while VT-RWKV aligns visual concepts with textual concepts. Lesion concepts serve as intermediate bottlenecks for prediction through a concept-to-grade chain, with support for test-time physician intervention. Finally, the paper introduces a dual uncertainty-aware fusion module based on "concept uncertainty + grading uncertainty," accompanied by a generalization bound analysis demonstrating the superiority of dynamic fusion over static fusion. Experiments on MFIDDR and DRTiD demonstrate that the proposed method outperforms several multi-view baselines in DR grading and lesion concept recognition, while maintaining favorable inference efficiency.

**Compliance With Llm Reviewing Policy:**

Affirmed.

**Final Justification:**

see `Rebuttal Acknowledgement`

**Key Questions For Authors:**

1. Theorem 1 / Corollary 1 requires dynamic weights to be negatively correlated with prediction loss and concept loss. Can you prove that the specific dual-uncertainty weights given in Eq. (15) induce this negative correlation under your training objective, rather than just exhibiting empirical correlation on the test set?
2. The paper frequently employs these terms. Please define precisely what "causal" and "verification" mean in your context, and explain how they differ from general concept bottleneck prediction chains, uncertainty estimation, and post-hoc reporting.
3. The textual descriptions are globally shared, GPT-generated, and physician-verified static knowledge. Please provide stricter ablations: without text (concept label supervision only); randomized/shuffled text; human-simplified vs. GPT text.
4. Your method uses lesion masks, concept labels, and doctor-verified GPT text as extra information. Are comparisons with baselines controlled for these additional supervisory signals? If not, can you provide fair comparisons under the "same-supervision budget"?
5. Beyond correlation with loss, can you report calibration metrics (ECE, Brier), risk-coverage curves, rejection performance, or clinically safety-relevant analyses?

**Limitations:**

The authors should explicitly discuss the following limitations:

1. Heavy additional supervision: Concept labels, external segmentation models, and physician-verified text all increase deployment costs.

2. Within-dataset validation does not represent cross-domain clinical applicability; external generalization risks should be explicitly acknowledged.

3. GPT-generated text, even when verified, may introduce knowledge biases or overly standardized descriptions, diminishing individual case variations.

4. "Interpretability" does not equate to "causal correctness" or "clinical safety"; overstated claims should be avoided.

5. Erroneous confidence and automation bias in medical scenarios may lead to misdiagnosis risks; failure modes and human-machine collaboration boundaries should be supplemented.

**Strengths And Weaknesses:**

## Strengths

1. Rather than making isolated improvements, the paper attempts to systematically address "how to make multi-view DR models more trustworthy." The end-to-end organization from input to reasoning to verification is clear, and in the context of medical AI, this holds more practical significance than purely pursuing accuracy.

2. The paper integrates multi-view CBM, visual-textual concept alignment, dual-uncertainty fusion, and physician intervention within a unified framework, achieving relatively high engineering integration. Hilbert-RWKV is employed for local continuity preservation, VT-RWKV for cross-modal enhancement at the concept level, and DU-MVFD for fusion, with relatively clear module responsibilities.

3. On two public datasets, the method achieves consistent improvements in both grading and concept prediction; additionally, module ablations, scanning strategy comparisons, intervention experiments, and certain efficiency analyses are provided. Compared to many medical papers that only present final classification performance, this submission offers richer experimental dimensions.

4. Performance gains from test-time concept correction demonstrate that intermediate concepts are not merely decorative explanation layers, but play substantive roles in final decision-making. This point is crucial for concept bottleneck-type work.

## Weaknesses
1. The theoretical conclusion relies on negative correlation between dynamic weights and losses, yet the paper only provides empirical scatter plots rather than proving this property holds for the designed weight formulation. The "provenance-enabled verification" claim is overstated; the implementation amounts to uncertainty-based weighting with concept traceability, lacking strict provenance tracking or audit mechanisms. Uncertainty calibration metrics (ECE, Brier scores) are absent. External supervision sources (segmentation masks, GPT-generated text) introduce potential biases insufficiently analyzed.

2. "Causal reasoning" and "for the first time" are overclaimed without causal identification analysis or strict novelty verification. Experimental transparency is lacking regarding text prompt contributions and baseline adaptation fairness.

3. Evidence remains dataset-bound. Cross-device, cross-center, or cross-population validation is absent. Comparison with recent retinal foundation models or medical VLMs is insufficient. Physician intervention experiments are offline simulations rather than real workflow evaluations.

4. Individual modules (Hilbert-RWKV, CBM, evidential uncertainty, text alignment) are not novel; the contribution lies primarily in task-specific integration. The explicit incorporation of concept uncertainty into multi-view fusion with theoretical motivation is the most distinctive element, though the theory-algorithm closure is incomplete.

---

> ### Author Rebuttal · Authors · 2026-03-31
>
> 1. **Proof & Validation of the Positive Correlation Between Dual-Uncertainty and Grading Loss:**
>
> **Proof:** For cross-entropy loss, $\ell = -\sum_{i=1}^{N} y_i \log p_i$. Let the true class be $t$. Since $y_t=1$ and $y_i=0$ for $i\neq t$, we have $\ell = -\log p_t = -\log p_{\text{true}}$. Then $\frac{d\ell}{dp_{\text{true}}} = -\frac{1}{p_{\text{true}}} < 0$ for $p_{\text{true}} \in (0,1]$. Therefore, $\ell$ is a monotonically decreasing function of $p_{\text{true}}$, which implies $\mathrm{Cov}(p_{\text{true}}, \ell) < 0$. Hence, the mono-confidence $p_{\text{true}}$ is negatively correlated with the loss. According to the property of uncertainty, it follows that $$
> \frac{\partial \Psi_{v,c_j}^{\mathrm{con}}}{\partial p_{v,c_j}^{\mathrm{con}}} < 0 \, \frac{\partial \Psi_{v}^{\mathrm{gr}}}{\partial p_{v}^{\mathrm{gr}}} < 0, \quad \forall v, c_j$$ Since the grading and concept uncertainties of each view are negatively correlated with $p_{v}^{\mathrm{gr}}, p_{v,c_j}^{\mathrm{con}}$ , the conclusion can be obtained.
>
> **Validation:** We show the ECE of dual-uncertainty fusion. Results in Tables R1 and R2 show that uncertainty calibration reduces ECE, indicating reliability.
>
> **Table R1. Comparison of ECE before and after grading uncertainty calibration. (%)**
>
> |Uncalibrated ECE|Calibrated ECE|ΔECE|
> |:--- |:--- |:---|
> |8.21|5.01|↓3.20|
>
> **Table R2. Comparison of ECE before and after concept uncertainty calibration. (%)**
>
> |Uncalibrated ECE | Calibrated ECE | ΔECE |
>  |:--- |:--- |:--- |
> |6.05|4.09|↓1.96|
>
> 2. **Modules Novelty:** We believe it is inappropriate to overlook the novelty brought by the integration of dual-uncertainty fusion, Hilbert-RWKV, and a CBM specifically designed for the multi-view setting. In particular, existing CBM variants largely lack systematic study in multi-view scenarios and have not analyzed the relation between concept loss and outcome generalization loss. Moreover, introducing the Hilbert curve into the RWKV framework is the first attempt. Perhaps this is because the related work section in our paper was not detailed, and we will revise it.
>
> 3. **Experimental Fairness:** In Table 3 of the paper, we presented the results without text or mask. Here, we show them again for clarity. In response to your suggestion, we supplement the manuscript with descriptions and experiments in both text and mask. Importantly, all models are evaluated under a unified experimental protocol, including the same loss function, hyperparameters, batch, epochs, and data split. Among all methods, Multi-Task, MVCBM, CEM, PCBM, SSMVCBM, and our method have concept supervisions, whereas the others do not. To ensure fairness, we report the results without mask or text in Tables R3-R5.
>
> **Table R3. Comparison w/o masks/concepts (consistent with our settings).**
> |Method|Mask|Concept|
> |---|---|---|
> |CrossFit|✗|✗|
> |ETMC|✗|✗|
> |MVCINN|✗|✗|
> |CVSRA-ViT|✗|✗|
> |SMVDR|✓|✗|
> |WMIMVDR |✓|✗|
> |Multi-Task|✓|✓|
> |MVCBM |✓ |✓|
> |CEM|✓|✓|
> |PCBM|✓|✓|
> |SSMVCBM|✓|✓|
> |CLAT| ✗ |✓|
> |Ours|✓|✓|
>
> **Table R4. Ablation on Text. (%)**
>
> |Setting|Grading Acc|Concept F1|
> |---|---:|---:|
> |no text|85.68|64.39|
> |disordered|83.57|63.32|
> |simplified| 86.03 | 66.90 |
> |ours|**86.75**|**68.43**|
>
> **Table R5. Ablation on Mask. (%)**
>
> |Method|Grading Acc↑|Concept F1↑|
> |---|---:|---:|
> |no text&mask|84.94|65.89|
> |no mask|85.21|66.26|
> |ours|**86.75**|**68.92**|
>
> 4. **Provenance**
>
> For provenance, our framework follows an interpretable reasoning chain: multi-view images → mask-highlighted suspicious regions → lesion concepts → final diagnosis. Textual knowledge only describes lesion appearance and its diagnostic implications, while the core reasoning is realized through a concept bottleneck model consistent with ophthalmologists’ practice. Therefore, our method is an interpretable concept-guided vision model rather than a mere textual explanation, and the concept layer remains manipulable for human intervention. This kind of reasoning paradigm has also been recognized in clinical practice [1,2].
>
> 5. **Out-of-Domain(OOD)**
>
> As requested, we add OOD experiments.
>
> **Table R4. OOD DR Grading Performance Comparison on DRTiD (Trained on MFIDDR). (%)**
> |Method|Acc|F1|
> |---|---:|---:|
> |MVCINN|28.67|65.26|
> |SMVDR|78.55|78.11|
> |WMIMVDR|78.04|77.74|
> |MVCBM| 76.46|76.89|
> |CEM|78.62|77.32|
> |Ours|**79.31**|**79.02**|
>
> 6. **Limitations**
>
> We acknowledge your concerns that masks may increase deployment cost and that causal correctness is not equivalent to interpretability. We will revise the wording accordingly. Moreover, the text in our model is clinician-verified knowledge rather than unchecked output. We will further improve doctor-interactive concept intervention in future work, especially regarding failure modes and human–machine collaboration boundaries.
>
> [1]Transparent medical image AI via an image–text foundation model grounded in medical literature, Nature medicine, 2024.
>
> [2]Transparency of medical artificial intelligence systems, Nature Reviews Bioengineering, 2026.

---

> > ### Author Rebuttal · Reviewer_YJf7 · 2026-04-02
> >
> > The authors have substantially weakened the experimental concerns through solid supplementary experiments, sufficient to support the conclusion that "the method works." However, the theoretical part only provides explanations at the level of mathematical definitions, failing to establish a rigorous theoretical closure between the training objective and the generalization bound conditions. If the authors revise the terminology as promised (by weakening "causal"/"provenance") and supplement the requested failure-mode discussions, I believe this would be much improved. I would raise my score for this work.

---

> > > ### Author Response · Authors · 2026-04-07
> > >
> > > Thank you very much for your positive and constructive feedback. We sincerely appreciate your recognition that the additional experiments have substantially addressed the experimental concerns and are sufficient to support the conclusion that the method works. We also appreciate your valuable suggestions on the theoretical presentation and terminology. Following your advice, we will further refine the wording by weakening terms such as “causal” and “provenance,” and we will add the requested discussion on failure modes to make the paper clearer and more rigorous.

---

### Official Review · Reviewer_Kf9r · 2026-03-11

**Soundness:** 3
**Presentation:** 3
**Significance:** 2
**Originality:** 3
**Overall Recommendation:** 3
**Confidence:** 3

**Summary:**

This paper addresses diabetic retinopathy prediction with a fully interpretable diagnostic framework that brings in multiple data sources, incorporates interpretable causal reasoning, and a verifiable results evaluation. They bring in additional data (multimodal rather than unimodal) by using a LLM that extracts information from patient notes, additional images, and lesion masks. Reasoning is achieved via lesion concepts that are incorporated into their framework.

**Compliance With Llm Reviewing Policy:**

Affirmed.

**Final Justification:**

My score remains unchanged. The rebuttal validated my concerns by showing that the different ablation studies didn't have significantly different outcomes.

**Key Questions For Authors:**

Are the performance results truly different between the groups? This could be clarified by bootstrapping using the test data and assigning confidence intervals to the results.

Given the results of the ablation study, are all the modules necessary? Could this be simplified further?

**Limitations:**

No, there are no limitations discussed. One limitation here is the need for a lot of data to be pulled for test time (e.g, in the real world, you may have the retinal images but not the masks available or the clinical notes). Another limitation is the lack of any other external test sets.

**Strengths And Weaknesses:**

Strengths: I really like the addition of explainability to the models and the dynamic fusion of multiple images. I think the approach to developing this framework is interesting and leverages two key things: multi-modality and explainability.

Weakness: Looking at table 3 and the ablations, it seems like several of the modules such DU-MVFD, Hilbert-RWKV, VT-RWKV added much to the performance accuracy. I would like to see some bootstrapping done in the ablation studies to get confidence intervals and be able to tell what is truly significantly different. I would say the same issues appear in Tables 1, 4, and 5: unclear that the author's methods is truly an improvement. So while I think the methods are cool, I'm not entirely convinced without some bootstrapping of the test data.

---

> ### Author Rebuttal · Authors · 2026-03-30
>
> We sincerely appreciate your recognition of our method. In response to your concern that the ablation studies did not include bootstrapping to estimate confidence intervals, we carry out an additional six days of experiments and provide the following response.
>
> **Q1&W:** We perform three independent random splits of the MFIDDR (34,452 images) and DRTiD (3100 images) datasets into training, validation, and test sets with a 7:1:2 ratio, consistent with the original experimental settings [1–3]. We then conduct extensive experiments on a single NVIDIA RTX 4090 GPU. The detailed results are provided in the table below:
>
> **Table 1. Comparison of diagnosis accuracy and concept AUPR across different methods on the MFIDDR and DRTiD datasets.**
> | Dataset | Metric | ETMC | MVCINN | SMVDR | Multi-Task | MVCBM | CEM (MV) | PCBM (MV) | SSMVCBM | CLAT (MV) | ProConMV (ours) |
> |---------|--------|------|--------|-------|------------|-------|----------|-----------|----------|-----------|-----------------|
> | MFIDDR | Diagnosis Accuracy | 81.27±1.34 | 80.43±2.26 | 83.78±2.01 | 83.56±1.18 | 83.41±1.08 | 84.36±0.94 | 83.41±1.09 | 82.58±1.16 | 83.14±1.11 | 86.52±0.73 |
> | DRTiD | Diagnosis Accuracy | 65.71±1.47 | 67.94±3.38 | 74.81±1.92 | 72.96±1.13 | 71.82±1.21 | 74.38±0.88 | 74.65±0.95 | 73.74±2.02 | 74.69±0.97 | 76.63±0.69 |
> | MFIDDR | Concept AUPR |------------|------------ |------------| 54.83±1.12 | 61.71±0.96 | 65.32±0.88 | 68.27±0.81 | 66.14±0.93 | 64.02±1.07 | 72.14±0.72|
> | DRTiD | Concept AUPR | ------------| ------------ |------------| 47.18±1.09 | 48.73±1.03 | 48.61±0.97 | 52.74±0.86 | 53.37±0.91 | 51.96±0.95 | 55.68±0.76 |
>
> As shown in Table 1, ProConMV achieves the best overall performance on both datasets, outperforming all competing methods in both diagnosis accuracy and concept AUPR. These results demonstrate the effectiveness and robustness of our method across different multi-view DR benchmarks.
>
> **Q2&W:** In accordance with your suggestion, we re-run and simplify part of the ablation experiments under the Q2&W setting. And we will complete the reconstruction of all ablation experiments in the follow-up revision. The results are as follows：
>
> **Table 2. Bootstrapped mean ± standard deviation of the ablation results on grading and concept prediction performance on MFIDDR.**
>
> | Setting | Grading Acc | Kappa | Concept AUPR | F1 |
> |---|---:|---:|---:|---:|
> | Baseline + Mask | 83.41±1.08 | 69.37±1.01 | 61.71±0.96 | 60.01±1.06 |
> | w/o DU-MVFD | 86.08±0.74 | 75.01±0.69 | 70.52±0.83 | 67.74±0.78 |
> | w/o Text, VT-RWKV | 84.92±0.86 | 74.26±0.81 | 68.70±1.43 | 64.34±0.91 |
> | w/o Hilbert-RWKV | 85.64±0.93 | 72.84±0.94 | 67.93±0.87 | 66.31±0.83 |
> | w/o Mask | 85.36±0.85 | 74.12±0.80 | 68.74±0.92 | 66.45±0.88 |
> | **ProConMV (ours)**| **86.52±0.73** | **75.92±0.56** | **72.14±0.72** | **68.31±0.61** |
>
>
> **Table 3. Bootstrapped mean ± standard deviation comparison of different backbones on the MFIDDR dataset.**
>
> | Backbone | Grading Acc | Grading F1 | Concept Pred. AUPR | Concept Pred. F1 |
> |---|---:|---:|---:|---:|
> | VGG-16 | 85.48±0.96 | 85.11±0.88 | 67.42±0.91 | 66.73±0.84 |
> | ResNet-50 | 85.64±0.89 | 83.92±0.95 | 67.58±0.87 | 65.94±0.92 |
> | ViT-B | 85.97±0.81 | 85.29±0.86 | 55.23±1.04 | 53.06±0.98 |
> | Swin v1-S | 85.94±0.77 | 85.88±0.79 | 66.81±0.93 | 60.34±0.89 |
> | Swin v2-S | 86.21±0.74 | 85.62±0.82 | 61.73±0.97 | 57.76±0.91 |
> | VMamba | 83.26±1.08 | 82.47±1.02 | 59.18±0.95 | 52.43±1.06 |
> | **Hilbert-RWKV** |**86.52±0.73** | **86.27±0.58** | **72.14±0.72** | **68.31±0.61** |
>
> Table 1 and 3 shows that all key components, including Mask, Hilbert-RWKV, VT-RWKV, and DU-MVFD, consistently improve the final performance. Notably, compared with other small-scale multi-view DR datasets, ablation experiments conducted on MFIDDR, a large-scale medical dataset with 34,452 images, provide higher credibility.
>
> **Limitation 1:** We acknowledge the concern about the complexity of using masks and clinical notes. However, our clinical notes are diagnosis-guided clinical priors rather than physician reports (diagnostic answers), making them easier to obtain and encode, and no information leakage. The mask preliminarily highlights fundus regions of interest and enhances image features (analogous to enhancement using SAM). The low inference time reported in our paper also suggests that this design introduces limited complexity.
>
> **Limitation 2:** We acknowledge your concern regarding the lack of external datasets. It is worth noting that our model is extensible to other multi-view medical domains, such as skin cancer and chest disease analysis. Our paper will add experimental results on skin or chest diagnosis in the appendix.
>
> [1] MVCINN: multi-view diabetic retinopathy detection using a deep cross-interaction neural network, AAAI 2023
>
> [2] Cross-field transformer for diabetic retinopathy grading on two-field fundus images, BIBM 2022
>
> [3] Qmix: Quality-aware learning with mixed noise for robust retinal disease diagnosis, TMI 2025

---

> > ### Author Rebuttal · Reviewer_Kf9r · 2026-04-07
> >
> > The bootstrapped ablation studies show significant overlap in results-- particularly on grading accuracy. Thank you to the authors for their response, but my score will stand.

---

> > > ### Author Response · Authors · 2026-04-07
> > >
> > > **Table 4. Grading and concept experimental results under three dataset splits on the MFIDDR dataset. (using the same splits as Tables 2 and 5)**
> > >
> > > | Setting | Grading Acc | Grading Kappa | Concept AUPR | F1 |
> > > |---|---|---|---|---|
> > > | &nbsp;Baseline + Mask&nbsp; | &nbsp;83.22 / 82.44 / 84.57&nbsp; | &nbsp;69.12 / 68.51 / 70.48&nbsp; | &nbsp;61.56 / 60.83 / 62.74&nbsp; | &nbsp;59.10 / 59.76 / 61.17&nbsp; |
> > > | &nbsp;w/o DU-MVFD&nbsp; | &nbsp;86.23 / 85.28 / 86.73&nbsp; | &nbsp;75.23 / 74.24 / 75.56&nbsp; | &nbsp;70.69 / 69.62 / 71.25&nbsp; | &nbsp;67.89 / 66.90 / 68.43&nbsp; |
> > > | &nbsp;w/o Text, VT-RWKV&nbsp; | &nbsp;85.68 / 83.99 / 85.09&nbsp; | &nbsp;74.43 / 73.38 / 74.97&nbsp; | &nbsp;70.12 / 67.26 / 68.72&nbsp; | &nbsp;64.39 / 63.41 / 65.22&nbsp; |
> > > | &nbsp;w/o Hilbert-RWKV&nbsp; | &nbsp;85.72 / 84.67 / 86.53&nbsp; | &nbsp;72.68 / 71.99 / 73.85&nbsp; | &nbsp;67.71 / 67.19 / 68.89&nbsp; | &nbsp;66.12 / 65.59 / 67.22&nbsp; |
> > > | &nbsp;w/o Mask&nbsp; | &nbsp;85.21 / 84.59 / 86.28&nbsp; | &nbsp;73.98 / 73.40 / 74.98&nbsp; | &nbsp;68.59 / 67.90 / 69.73&nbsp; | &nbsp;66.26 / 65.68 / 67.41&nbsp; |
> > > | **&nbsp;ProConMV (ours)&nbsp;** | **&nbsp;86.75 / 85.70 / 87.11&nbsp;** | **&nbsp;76.05 / 75.31 / 76.40&nbsp;** | **&nbsp;72.26 / 71.37 / 72.79&nbsp;** | **&nbsp;68.43 / 67.65 / 68.85&nbsp;** |
> > >
> > >
> > > **Table 5. Bootstrapped mean ± standard deviation of the ablation results on grading and concept prediction performance on MFIDDR. (This can be compared with the above Table 2 and Table 4.)**
> > >
> > > | Setting | Grading Acc | Grading Kappa | Grading Specificity | Grading F1 | Concept AUPR | Concept F1 |
> > > | :-- | :--: | :--: | :--: | :--: | :--: | :--: |
> > > | Baseline + Mask | 83.41±1.08 | 69.37±1.01 | 89.42±0.48 | 81.24±1.51 | 61.71±0.96 | 60.01±1.06 |
> > > | w/o DU-MVFD | *86.08±0.74* | *75.01±0.69* | *90.68±0.47* | *84.73±0.56* | *70.52±0.83* | *67.74±0.78* |
> > > | w/o Text, VT-RWKV | 84.92±0.86 | 74.26±0.81 | 89.95±1.48 | 83.16±1.32 | 68.70±1.43 | 64.34±0.91 |
> > > | w/o Hilbert-RWKV | 85.64±0.93 | 72.84±0.94 | 91.27±1.05 | 84.05±1.44 | 67.93±0.87 | 66.31±0.83 |
> > > | w/o Mask | 85.36±0.85 | 74.12±0.80 | 90.11±1.29 | 82.92±2.39 | 68.74±0.92 | 66.45±0.88 |
> > > | **ProConMV (ours)** | **86.52±0.73** | **75.92±0.56** | **92.79±0.82** | **86.35±1.63** | **72.14±0.72** | **68.31±0.61** |
> > >
> > > **Table 6. Grade Distribution of the DR dataset MFIDDR.**
> > >
> > > | Grade | Count | Percentage |
> > > |---|---:|---:|
> > > | 0 | 20,992 | 60.93% |
> > > | 1 | 7,328 | 21.27% |
> > > | 2 | 3,008 | 8.73% |
> > > | 3 | 2,484 | 7.21% |
> > > | 4 | 640 | 1.86% |
> > >
> > > 1. We appreciate the reviewer’s observation regarding the statistical significance of our ablation results. We believe the reviewer’s observation of overlapping results mainly arises from **the class-imbalanced nature of the DR dataset**. **Under such an imbalance, different random splits can noticeably affect test accuracy, with fluctuations of about 0.5%–1.5%. Therefore, a direct vertical comparison between ablation results obtained from more favorable and less favorable data splits is not fair.** It is the root cause of the overlap within the standard deviation ranges that you pointed out. A more appropriate evaluation is to **examine whether the full model consistently outperforms its partial variants under each independent split**. We acknowledge that our rebuttal did not present this point clearly enough. To address this, we now **report the ablation results under all three independent splits in Tables 4 and 5**, which more fairly demonstrate the consistent advantage of the full model.
> > >
> > > 2. Moreover, to facilitate a more comprehensive evaluation, we have incorporated Grading F1-score and Specificity into our reporting. Importantly, the DR grading dataset is highly imbalanced. Specificity, Kappa, and F1-score are more robust and clinically relevant for evaluating imbalanced grading. Our results demonstrate that the full model not only achieves the highest mean Grading Accuracy but also consistently outperforms all ablated variants in Grading Kappa, Grading Specificity, and Grading F1, suggesting that these gains are systemic rather than the result of random variation.
> > >
> > > 3. Our primary objective is to optimize both clinical grading and interpretable concept prediction. The superior results in Concept AUPR and Concept F1 indicate that the proposed modules effectively enhance discriminability and consistency at the conceptual level. Therefore, we believe the ablation results should be interpreted as demonstrating stable, comprehensive gains across both grading and concept tasks, rather than being judged solely by the statistical separation of a single metric.
> > >
> > > If you review our second-round response and find that it clarifies your questions, we would sincerely appreciate your reconsideration of our work.

---

### Official Review · Reviewer_t6NQ · 2026-03-13

**Soundness:** 3
**Presentation:** 3
**Significance:** 3
**Originality:** 3
**Overall Recommendation:** 4
**Confidence:** 3

**Summary:**

The authors present ProConMV, a concept-driven framework designed for transparent multi-view diabetic retinopathy (DR) identification through provenance tracking. Key innovations include a Hilbert-RWKV backbone for enhanced spatial feature extraction, a VT-RWKV module for mapping visual lesion data to clinical text, and an uncertainty-informed fusion mechanism that balances different views based on concept and grading reliability. Evaluations using the MFIDDR and DRTiD datasets demonstrate consistent outperformance in classification and grading, supported by clinical intervention simulations and computational efficiency studies.

**Compliance With Llm Reviewing Policy:**

Affirmed.

**Final Justification:**

I increased my score to 4, given the authors' rebuttal efforts.

**Key Questions For Authors:**

Can you provide the exact textual prompts/descriptions used (and their verification protocol), and commit to releasing them for reproducibility? Also clarify the “GPT-5” versus the GPT-4 citation inconsistency.

Could you formalize how “clinical guidelines” inform the causal chain (e.g., explicit rules/graphs, constraints on the grade decoder)? Are there failure cases where guideline priors conflict with data?

How were backbone baselines trained to ensure fairness (hyperparameters, data augmentations, early stopping)? Given large parameter differences, can you report FLOPs and match training budgets?

**Limitations:**

Yes.

**Strengths And Weaknesses:**

Strength

Strong empirical results + efficiency on relevant multi-view DR benchmarks. The draft evaluates on MFIDDR and DRTiD with both grading and lesion-concept prediction, reporting consistent improvements over a broad set of multi-view baselines and competitive inference time.

Clear architectural novelty in the vision backbone. The proposed Hilbert-RWKV combines RWKV-style linear attention with Hilbert-curve–based token ordering / multi-directional attention to better preserve spatial locality while keeping linear complexity, and the paper backs this up with backbone + scanning-strategy ablations.

Interpretable + intervenable pipeline rather than post-hoc explanation. The method explicitly models lesion concepts, aligns them with text via a VT-RWKV block, and uses concept/grading uncertainties for fusion; the draft also includes an intervention mechanism where correcting concepts/views improves the final decision, alongside qualitative “step-by-step” inference visualizations.

Weakness

Multimodal/provenance claims rely on derived, non-patient-specific signals. The “text modality” appears to be LLM-generated, guideline-style lesion descriptions (shared templates), and the “mask modality” is produced by an external pretrained segmentation model, so the method risks reading as feature engineering with frozen priors rather than leveraging truly independent patient-native modalities.

Uncertainty-aware fusion is not validated as calibration. The paper motivates fusion weights via uncertainty and generalization arguments, but the empirical evidence is mainly correlation-style plots; it lacks standard calibration evaluation (e.g., reliability diagrams, ECE/ACE, post-hoc calibration baselines) to justify uncertainty as trustworthy confidence.

Interpretability and clinical usefulness remain speculative. The “explanations” are largely generated from fixed text templates rather than clinician-authored notes, with no clinician study or decision-support evaluation.

---

> ### Author Rebuttal · Authors · 2026-03-30
>
> Thank you for acknowledging the strengths of our study. We are grateful for your recognition and thoughtful feedback, which inspires us to continue our efforts.
>
> W1&W3:
> 1) We fully agree with your point that our textual modality should be regarded as prior knowledge. The model uses textual priors to guide reasoning and enhance feature learning. We do not use clinical reports because they contain explicit diagnostic conclusions, and using them as input would cause severe information leakage and deviate from the practical setting of clinical diagnosis. The generated masks highlight diagnostically relevant regions and act as a form of information enhancement derived from pretrained models (e.g., SAM). It is similar to how clinicians in real practice focus first on lesion regions and diagnostically relevant abnormalities.
>
> 2) Regarding provenance, our interpretation is as follows. The reasoning process in our framework can be summarized as: multi-view fundus images → suspicious regions highlighted by masks → medical concepts (lesions) → final diagnosis. The purpose of this textual information from the diagnostic guidelines is to describe the manifestations of the pathological concepts and the possible diagnostic outcomes they may lead to. This reasoning chain, built on a concept bottleneck model, is aligned with the way ophthalmologists perform diagnosis in practice (and can be supported by clinician agreement protocols). Moreover, similar explainable concept-guided reasoning paradigms have been validated in prior studies [1,2]. Hence, our multi-view concept reasoning pipeline is an interpretable vision framework, rather than simply a textual explanation. Moreover, concept information serves as an intermediate and manipulable neuron, through which human intervention can influence the diagnostic outcome.
>
> W2: We further report the Expected Calibration Error (ECE) results of dual-uncertainty fusion. The results in Tables 1 and 2 show that uncertainty calibration consistently reduces ECE across all confidence intervals, indicating improved reliability. Notably, grading uncertainty calibration yields larger overall gains, while concept uncertainty calibration shows particularly clear benefits in the high-confidence regime.
>
> **Table 1. Comparison of ECE before and after grading uncertainty calibration. (Unit: %)**
>
> | Confidence Interval | Uncalibrated ECE | Calibrated ECE | ΔECE |
> |:---|:---|:---|:---|
> |low [0.0, 0.4)|7.71|**3.38**|↓4.33|
> |mid [0.4, 0.7)|9.95|**6.78**|↓3.17|
> |high [0.7, 1.0]|8.02|**4.55**|↓3.47|
> |Full [0.0, 1.0]|8.21|**5.01**|↓3.20|
>
> **Table 2. Comparison of ECE before and after concept uncertainty calibration. (Unit: %)**
>
> | Confidence Interval | Uncalibrated ECE | Calibrated ECE | ΔECE |
> |:---|:---|:---|:---|
> |low [0.0, 0.4)|5.64|**4.02**|↓1.62|
> |mid [0.4, 0.7)|6.23|**4.40**|↓1.83|
> |high [0.7, 1.0]| 6.01|**3.68**|↓2.33|
> |Full [0.0, 1.0]| 6.05|**4.09**|↓1.96|
>
> Q1: Thank you for your suggestion. We already included representative examples of the text prompts and descriptions in Figure 2 of the original supplementary material, and we will add a complete version in the revised manuscript. GPT-5 came in August 2025, and a related paper appeared as a preprint in December 2025. We finished our manuscript during this period, so we did not include later updates to that paper. We will correct this error in the revised manuscript.
>
> Q2: For example, guideline-style descriptions explicitly characterize MA as small, scattered, red dot-like lesions and identify it as a hallmark lesion of Mild NPDR, which helps the model capture the corresponding lesion features and guides the grading decoder to follow a concept-to-grade reasoning path. However, we also observe rare error cases where EX or HE is detected, but the model still misclassifies the sample as a Mild NPDR stage.
>
> Q3: We initialize all backbones with ImageNet-pretrained weights and keep their default settings (e.g., ViT-Large). For fairness, we train all models with the same input resolution of 256 × 256, batch size of 8, and loss functions with identical parameter settings. We use Adam with an initial learning rate of 1e-5 and cosine annealing, train each model for 100 epochs without early stopping, and select the checkpoint with the highest validation disease accuracy for evaluation. Tables 3 and 4 below report the FLOPs of the main models.
>
> **Table 3. The FLOPs of different backbones.**
> | Backbone Model | FLOPs (G) |
> |:---|:---|
> |VGG16|80.43|
> |ViT-Large|312.53|
> |VMamba-Base|34.76|
> |Swin-v1-S|68.61|
> |**Hilbert-RWKV (ours)**|**26.15**|
>
> **Table 4. The FLOPs of the comparative models.**
> |Method|FLOPs (G)|
> |:---|:---|
> |WMIMVDR|29.47|
> |MVCINN|70.27|
> |SMVDR|72.61|
> |MVCBM|31.98|
> |SSMVCBM|37.86|
> |**ProConMV (ours)**|**26.15**|
>
> [1]Transparent medical image AI via an image–text foundation model grounded in medical literature, Nature medicine, 2024.
>
> [2]Transparency of medical artificial intelligence systems, Nature Reviews Bioengineering, 2026.

---

> > ### Author Rebuttal · Reviewer_t6NQ · 2026-04-06
> >
> > I still have reservations about the mask modality and interpretability.

---

> > > ### Author Response · Authors · 2026-04-06
> > >
> > > We greatly appreciate your response and are encouraged by your constructive feedback. We hope that our explanation thoroughly addresses your concerns.
> > >
> > > Our claimed interpretability does not suggest that the mere inclusion of masks or concept texts automatically makes the model interpretable. Instead, we decompose interpretability into three levels: **understandable inputs, traceable intermediate reasoning, and verifiable final decisions**. To understand the full framework, it is first necessary to recognize the interpretability of CBMs. Our method follows the paradigm of making predictions via explicit concept features, thereby preserving a transparent reasoning path: **image→concept→disease**.
> > >
> > > The first level is **understandable inputs**. Our point is not that we directly introduce textual medical records or diagnosis results, but rather that we enhance the input representation with human-understandable auxiliary information, including lesion masks and concept-aligned textual knowledge descriptions. The lesion masks explicitly highlight suspicious pathological regions, making the model’s visual focus more transparent, while the concept texts provide standardized clinical semantic descriptions for each lesion category. For example, guideline-style descriptions explicitly characterize MA as small, scattered, red dot-like lesions and identify it as a hallmark lesion of Mild NPDR. As a result, the model input is no longer just an opaque pixel space, but a more medically meaningful and structured representation that is easier for clinicians and patients to check. Following your suggestion, we will revise the description of **multimodal information** in the manuscript to instead refer to **input information** enhancement. Clinician-authored notes contain final conclusions and thus cannot be directly used for input.
> > >
> > >
> > > The most essential level of interpretability lies in the **traceable intermediate reasoning process**. Rather than directly mapping images to DR grades, our model first transforms each view into concepts with explicit medical meaning. Then it predicts the final grade through an explicit pathway. Through VT-RWKV, these visual concepts are aligned with clinical text descriptions, so that each concept representation reflects both the observed lesion features and their medical significance. For example, instead of directly outputting Moderate NPDR, the model first identifies concepts such as MA and HE, and then infers the grade based on their concept-level evidence and associated clinical knowledge. However, diagnostic reasoning is not always correct, particularly when concept results are inaccurate. In such cases, the intervention mechanism of CBMs becomes crucial. At test time, doctors can revise the concepts for a specific view, and the model will then re-infer its grade and update the final diagnosis accordingly. In our experiments, correcting all erroneous concepts improves grading accuracy to 91.56%, a 4.81% gain over the non-intervention setting. It proves that the interpretability of our model is not merely static visualization, but an interactive mechanism that enables experts to refine the model’s decision through concept-level intervention. **Here, we clarify the interpretability of both the concept reasoning process, and also point out that there exist a few cases where the model’s reasoning may conflict with diagnostic guidelines.**
> > >
> > > The third level is **verifiable final decisions**. Our model not only outputs a grade for each view but also estimates both concept uncertainty and grading uncertainty for each view, which determines the fusion weights. If a view is unreliable at the concept or grading level, its contribution to the final diagnosis is correspondingly reduced. As a result, the final prediction not only indicates what the diagnosis is, but also makes it possible to trace which view and which concept reasoning path contributed most, as well as how reliable those contributions are. **Here, we liken this process to how clinicians explicitly judge different views before reaching a comprehensive diagnosis.**
> > >
> > > |Reasoning stage | Description|
> > > |---|---|
> > > |**1. Understandable input** | The model takes **images, lesion masks (regions of interest, ROI), and standardized clinical concept text** as input, converting black-box pixels into structured medical evidence.|
> > > |**2. Concept Extraction**|For each view, the model predicts meaningful lesion concepts (e.g., **MA, HE**) with confidence scores. |
> > > |**3. Single-view Grading** | Each view independently predicts DR grade probabilities and estimates both **concept and grading uncertainty**. |
> > > |**4. Multi-view Fusion** | The final diagnosis is obtained through **uncertainty-aware fusion**, where more reliable views receive higher weights, making the decision traceable and evidence-based.|
> > > |**5. Report Generation** | The model summarizes lesion evidence, lesion regions, per-view grades, and diagnostic evidence into a structured clinical report.|

---

### Decision · Program_Chairs · 2026-04-30

**Decision:**

Accept (regular)

**Comment:**

There is a consensus among reviewers about the novelty of the method (e.g., Hilbert-RWKV architecture and the dual uncertainty-aware fusion guided by generalization bounds), and the strong empirical performance of the model. There were initial questions about missing ECE, theoretical proofs, external validations, implementation details, etc. The authors have fully (3 reviewers) and partially (1 reviewer) resolved the issues through rebuttal and follow-up discussion. For this reason, the AC recommends Accept for this submission.